# Prediction-Powered Semi-Supervised Learning with Online Power Tuning

**Noa Shoham[1]   Ron Dorfman[1]   Shalev Shear[1]   Kfir Y. Levy[1]   Yaniv Romano[1,2]**
[1]Department of Electrical and Computer Engineering, Technion IIT
[2]Department of Computer Science, Technion IIT

## Abstract

Prediction-Powered Inference (PPI) is a recently proposed *statistical inference* technique for parameter estimation that leverages pseudo-labels on both labeled and unlabeled data to construct an unbiased, low-variance estimator. In this work, we extend its core idea to semi-supervised learning (SSL) for *model training*, introducing a novel unbiased gradient estimator. This extension addresses a key challenge in SSL: while unlabeled data can improve model performance, its benefit heavily depends on the quality of pseudo-labels. Inaccurate pseudo-labels can introduce bias, leading to suboptimal models. To balance the contributions of labeled and pseudo-labeled data, we utilize an *interpolation parameter* and tune it on the fly, alongside the model parameters, using a one-dimensional online learning algorithm. We verify the practical advantage of our approach through experiments on both synthetic and real datasets, demonstrating improved performance over classic SSL baselines and PPI methods that tune the interpolation parameter offline.

## 1   Introduction

Semi-supervised learning (SSL) has garnered significant attention due to its success in leveraging both labeled and unlabeled data to improve model performance [1–13]. A standard SSL approach augments pseudo-labeled data with trusted labeled data and then fits a new model—or fine-tunes a pre-trained one—by minimizing empirical risk [4–6, 13–22]. A key limitation of this approach arises when the pseudo-labels are inaccurate. For instance, consider a minority subpopulation on which the pre-trained model performs poorly. In such cases, the resulting pseudo-labels introduce bias into the optimization objective. This issue is especially problematic when trusted labeled data is scarce and pseudo-labeled data is abundant. Under these conditions, the SSL model becomes biased toward the erroneous pseudo-labels, effectively ignoring the informative trusted labels from the minority group and failing to improve performance.

Recently, several approaches have been proposed to address the issue of biased risk using pseudo-labels, with Prediction-Powered Inference (PPI) [23–26] emerging as a notable example. PPI is designed to construct valid confidence intervals for parameters of interest by leveraging both labeled and pseudo-labeled data while correcting for potential biases in the latter, offering robustness in inference under minimal assumptions. Building on the principles introduced in PPI, subsequent works have extended these ideas to practical SSL settings [14, 27], incorporating corrections for pseudo-label bias directly into the learning objective to improve robustness and generalization.

Nevertheless, existing PPI-based SSL methods leave two critical gaps: **(i)** Their analyses evaluate the effect of pseudo-labels only asymptotically—at the **population optimum**, an idealized point never exactly reached during training—and offer no guarantees about how pseudo-labels affect the actual convergence of the learning algorithm. **(ii)** Realizing the full benefit of pseudo-labeled data hinges on an interpolation parameter that balances their quality against the variance of the labeled data. Prior work chooses this parameter offline, assuming knowledge of the pseudo-label quality and the labeled

39th Conference on Neural Information Processing Systems (NeurIPS 2025).

data variance [14, 24]. Unfortunately, these quantities are unknown in practice, so an estimated fixed value may be markedly sub-optimal.

**Contributions**

We present Prediction-Powered SSL (PP-SSL), a novel framework for unbiased semi-supervised learning. We analyze PPI-inspired gradient estimates and demonstrate how their variance decreases as the pre-trained teacher model's accuracy improves and the amount of unlabeled data increases. We further show how this reduced variance leads to faster convergence. This contrasts with prior work [14, 27], which provides only asymptotic guarantees and no finite-time convergence bounds.

Our work also guides us on how to choose the interpolation parameter $\lambda \in [0, 1]$ to optimally balance the effect of pseudo-labeled data against the variance of the labeled data. Unfortunately, the optimal choice of $\lambda$ relies on impractical prior knowledge of the labeled data variance and the teacher model's accuracy. To address this, we design an online learning approach that tunes $\lambda$ on the fly during training and achieves guarantees matching those of the optimal $\lambda$. This result highlights the significance of moving from offline to online analysis, enabling a rigorous account of training dynamics.

Numerical experiments on both real and synthetic data demonstrate the advantage of our online SSL framework on regression and classification tasks. Our method outperforms both classic (biased) SSL and PPI-like training methods in scenarios where there is a subgroup in the data on which the teacher model performs poorly. [1]

## 2 Preliminaries and background

Consider $n$ labeled data points, sampled i.i.d. from some unknown distribution $P = \mathbb{P}_X \times \mathbb{P}_{Y|X}$, denoted by $\mathcal{D}_l = \{(x^i, y^i)\}_{i=1}^n$, where $(x^i, y^i) \in \mathcal{X} \times \mathcal{Y}$. Also, assume we have $N$ unlabeled data points sampled i.i.d. from the same $\mathbb{P}_X$, denoted by $\mathcal{D}_{\text{unl}} = \{\tilde{x}^i\}_{i=1}^N$. In this work, we focus on the more common and interesting scenario where unlabeled data is abundant while labeled data is scarce and costly to obtain, i.e., $N \gg n$.

**Background and related work**

Semi-supervised learning (SSL) is a machine learning paradigm that leverages both labeled and unlabeled data to improve model performance. In contrast, standard supervised learning relies solely on labeled data and typically resort to optimizing the empirical loss (a.k.a. empirical risk):

$$L_n(w) := \frac{1}{n} \sum_{i=1}^n \ell(w; x^i, y^i), \tag{1}$$

where $w \in \mathbb{R}^d$ denotes the trainable model parameters. A common SSL approach is pseudo-labeling [4–6, 28], where a model $f : \mathcal{X} \to \mathcal{Y}$ is used to generate artificial labels for the unlabeled data to augment training. There are two main strategies for choosing this model. The first is self-training [14–17], where the same model being trained is also used to generate pseudo-labels that are continuously updated during training. In contrast, we adopt the teacher-student framework [18–22], in which a fixed teacher model produces pseudo-labels that remain constant while a separate student model is trained. This setup offers greater stability and is better suited for theoretical analysis of pseudo-label bias. The typical loss used in such settings is

$$L_{\text{SSL}}(w) = L_n(w) + \tilde{L}_N^f(w), \quad \text{where} \quad \tilde{L}_N^f(w) := \frac{1}{N} \sum_{i=1}^N \ell(w; \tilde{x}^i, f(\tilde{x}^i)) . \tag{2}$$

However, pseudo-labeling may introduce confirmation bias, where erroneous pseudo-labels reinforce the model's mistakes [5]. Theoretical analyses of pseudo-labeling in SSL have primarily focused on offline settings. Previous works [5, 17] study convergence and confirmation bias under confidence-based pseudo-labeling, but their analyses are limited to self-training with biased estimators. The authors of [22] analyze knowledge distillation (e.g., teacher-student) as a variance reduction mechanism but assume a self-distillation setup with identical teacher and student models

---

[1]Software for reproducing the experiments is available at https://github.com/noashoham/PP-SSL

and derive guarantees in an infeasible setting without ensuring unbiased risk estimation. In contrast, our theoretical contribution analyzes gradient variance in the online teacher-student setting and provides convergence guarantees under unbiased risk minimization.

In the context of statistical inference, Angelopoulos et al. [23] proposed the Prediction-Powered Inference (PPI) framework, which mitigates the bias introduced by pseudo-labels through their use on both labeled and unlabeled data, employing the following loss:

$$L_{\text{PPI}}(w) = L_n(w) + \tilde{L}_N^f(w) - L_n^f(w), \quad \text{where} \quad L_n^f(w) \coloneqq \frac{1}{n} \sum_{i=1}^n \ell(w; x^i, f(x^i)). \qquad (3)$$

This formulation reduces the estimator's variance when $f$ is accurate while remaining unbiased even when the teacher model $f$ is inaccurate. This is due to the fact that, in expectation, the pseudo-label loss on the unlabeled data equals the pseudo-label loss on the labeled data, thereby canceling out the bias introduced by the teacher. Since when teacher predictions are inaccurate PPI can result in higher variance (larger confidence intervals), Angelopoulos et al. [24] proposed `PPI++`, introducing a tuning parameter $\lambda \in [0, 1]$ to interpolate the standard, supervised loss with the PPI loss:

$$L_{\text{PPI++}}(w) = L_n(w) + \lambda \left( \tilde{L}_N^f(w) - L_n^f(w) \right) = (1 - \lambda) L_n(w) + \lambda L_{\text{PPI}}(w). \qquad (4)$$

The above generalizes the PPI loss, interpolating between supervised learning ($\lambda = 0$) and vanilla PPI ($\lambda = 1$). As the authors of [24] focus on parameter inference, they tune $\lambda$ offline to minimize the asymptotic variance of the parameter estimate. However, both PPI and `PPI++` focus exclusively on offline confidence interval construction for parameter estimation (e.g., linear model weights) and do not address semi-supervised training setting.

More recently, Doubly-Robust Self-Training [14] extended this line of work by applying a PPI-like loss for *training*, using a pre-defined step function for the tunable weight $\lambda$ and providing asymptotic variance guarantees. Sifaou and Simeone [27] similarly build on the PPI form, aiming to improve robustness to teacher model errors, but still restrict their analysis and application to offline settings.

In this work, we aim to extend the PPI framework to semi-supervised training by developing a method that adaptively tunes the interpolation weight $\lambda$ during training using an *online learning* algorithm. Our approach dynamically balances the contributions of labeled and pseudo-labeled data to minimize cumulative gradient variance, enabling more stable and effective learning than previous methods that rely on fixed or offline-tuned weight.

## 3 Prediction-Powered Semi-Supervised Learning

In this section, we describe our proposed Prediction-Powered SSL (`PP-SSL`) framework, an unbiased semi-supervised learning algorithm. `PP-SSL` introduces a novel training paradigm that leverages prediction-powered gradient estimates. Formally, our objective is to minimize the expected loss function $\mathcal{L} : \mathbb{R}^d \to \mathbb{R}$, defined as,

$$\min_{w \in \mathbb{R}^d} \mathcal{L}(w) \coloneqq \mathbb{E}_{(x,y) \sim P}[\ell(w; x, y)] .$$

To this end, we consider iterative first-order optimization approach, where in each round $t$, we have access to a batch of $n$ labeled samples, $\{(x_t^i, y_t^i)\}_{i=1}^n \sim P$, where $P = \mathbb{P}_X \times \mathbb{P}_{Y|X}$; a batch of $N$ unlabeled samples, $\{\tilde{x}_t^i\}_{i=1}^N \sim \mathbb{P}_X$; and a prediction model (i.e., teacher) $f : \mathcal{X} \to \mathcal{Y}$. Using the labeled data, the unlabeled data, and the teacher model, we compute a stochastic gradient estimate of $\mathcal{L}$, which is used to update the current iterate $w_t$. This process is repeated over multiple rounds, ultimately producing an output $w_{\text{out}}$. Since we focus on non-convex optimization problems (e.g., neural networks training), where global minimization is generally intractable, we aim to find an approximate stationary point for which $\|\nabla \mathcal{L}(w_{\text{out}})\|^2$ is near-zero in expectation.

**Teacher's quality.** The effectiveness of leveraging the teacher model depends on its ability to accurately predict the true labels. We quantify this through the teacher's prediction error, defined as

$$\mathcal{E}^f \coloneqq \mathbb{E}_{(x,y) \sim P}(y - f(x))^2 . \qquad (5)$$

As we will show later, $\mathcal{E}^f$ plays a central role in the convergence guarantees of our approach.

**Notations.** Throughout, we use $\|\cdot\|$ to denote the $L_2$ norm. We define $\mathcal{F}_t$ to be the filtration at iteration $t$, which encompasses all randomness up to that time. We use $\mathbb{E}$ and $\mathbb{V}$ to denote expectation and variance, respectively, with $\mathbb{E}_t$ and $\mathbb{V}_t$ representing their conditional counterparts given $\mathcal{F}_t$. Finally, we use standard big-O notation, where $\mathcal{O}(\cdot)$ hides numerical constants.

**Prediction-Powered gradients.** Our approach leverages stochastic gradient estimates inspired by the PPI++ loss (Equation (4)). Let us introduce the following notations (omitting iteration index $t$):

$$g^n := \frac{1}{n} \sum_{i=1}^n \nabla \ell(w; x^i, y^i); \quad g^{n,f} := \frac{1}{n} \sum_{i=1}^n \nabla \ell(w; x^i, f(x^i)); \quad \tilde{g}^{N,f} := \frac{1}{N} \sum_{i=1}^N \nabla \ell(w; \tilde{x}^i, f(\tilde{x}^i)) .$$

Here, $g^n$ represents a standard mini-batch gradient estimator based on $n$ labeled samples; $g^{n,f}$ replaces the true labels with predictions from the model $f$; and $\tilde{g}^{N,f}$ is the gradient based on the unlabeled samples with pseudo-labels generated by $f$. We use these gradients to construct a *prediction-powered gradient*, defined for some parameter $\lambda \in [0, 1]$ as follows:

$$g_{\text{PP}}^\lambda := g^n + \lambda \left( \tilde{g}^{N,f} - g^{n,f} \right) . \tag{6}$$

Since the labeled and unlabeled data are sampled from the same underlying feature distribution $\mathbb{P}_X$, it can be easily verified that $g_{\text{PP}}^\lambda$ is an unbiased gradient estimator:

$$\mathbb{E}\left[ g_{\text{PP}}^\lambda \right] = \mathbb{E}\left[ g^n \right] + \lambda \mathbb{E}\left[ \tilde{g}^{N,f} - g^{n,f} \right] = \mathbb{E}[g^n] = \nabla \mathcal{L}(w) . \tag{7}$$

Notably, the convergence of (unbiased) gradient-based optimization methods, such as stochastic gradient descent (SGD), that utilize this estimator depends on its variance. Intuitively, when we set $\lambda = 0$, we resort to standard SGD (with mini-batch gradients); however, as we show, if the predictions of $f$ are accurate, and with more unlabeled data, we can significantly reduce the gradient's variance, resulting in faster convergence for $\lambda > 0$.

We begin in Section 3.1 by analyzing the variance of the prediction-powered gradient and deriving the optimal tuning of $\lambda$ to minimize it. This tuning yields the tightest possible convergence bounds within our framework. However, the optimal value of $\lambda$ depends on unknown quantities, making this approach infeasible in practice. Therefore, in Section 3.2, we treat $\lambda$ as a trainable parameter and co-optimize it alongside the model parameters. Specifically, we employ a one-dimensional adaptive online learning algorithm to dynamically update $\lambda$ on the fly so as to minimize the cumulative variance of the gradients over time. We establish that this approach introduces only lower-order terms into the convergence rate.

**Assumptions.** For the purpose of formal analysis, we introduce the following assumptions. We assume the objective $\mathcal{L}$ is $\beta$-smooth, i.e., $\mathcal{L}(u) \leq \mathcal{L}(w) + \nabla \mathcal{L}(w)^\top (u - w) + \frac{\beta}{2} \|u - w\|^2$ for all $u, w \in \mathbb{R}^d$. In addition, suppose that the constants $\sigma^2$ and $\sigma_e^2$ defined below are finite:

$$\sigma^2 := \sup_{w \in \mathbb{R}^d} \mathbb{E}_{x,y \sim P} \|\nabla \ell(w; x, y) - \nabla \mathcal{L}(w)\|^2 < \infty , \tag{8}$$

and

$$\sigma_e^2 := \sup_{w \in \mathbb{R}^d} \mathbb{E}_{x,y \sim P} \left\| \nabla \ell_e^f(w; x, y) - \nabla \mathcal{L}_e^f(w) \right\|^2 < \infty , \tag{9}$$

where $\ell_e^f(w; x, y) := \ell(w; x, y) - \ell(w; x, f(x))$ is the instance-dependent *loss error function*, which quantifies the difference between the losses when using the true label $y$ and the teacher-provided pseudo-label $f(x)$, and $\mathcal{L}_e^f(w) := \mathbb{E}_{x,y \sim P}[\ell_e^f(w; x, y)]$ is the corresponding expected loss error function. Note that the assumption in Equation (8) is standard and establishes a bounded variance of the instance-dependent loss gradient. The assumption in Equation (9) implies that the variance of the noisy loss error gradient is bounded as well. The next lemma establishes that $\sigma_e^2$ can be directly related to the teacher's expected prediction error $\mathcal{E}^f$ defined in Equation (5).

**Lemma 3.1.** *Assume that the gradient $\nabla \ell(w; x, y)$ is $L_Y$-Lipschitz in $y$, i.e., for any $w \in \mathbb{R}^d, x \in \mathcal{X}$, and $y_1, y_2 \in \mathcal{Y}$, we have $\|\nabla \ell(w; x, y_1) - \nabla \ell(w; x, y_2)\| \leq L_Y |y_1 - y_2|$. Then, $\sigma_e^2 \leq L_Y^2 \cdot \mathcal{E}^f$.*

In Appendix A, we provide a proof and show that the Lipschitzness condition applies to both the squared and the logistic losses, among others. In the next section, we show how the performance of our approach depends on $\sigma_e^2$, which is related to $\mathcal{E}^f$ by the above inequality.

## 3.1 Prediction-Powered SSL with optimal tuning

Next, we analyze the variance of the gradient estimates defined in Equation (6) and show that, with an optimal choice of $\lambda$, the variance can be significantly lower than that of the standard gradient estimates based solely on labeled data.

**Lemma 3.2.** *For every $\lambda \in \mathbb{R}$, the variance of $g_{PP}^{\lambda}$ defined in Equation* (6) *is bounded as follows:*

$$\frac{n}{4} \cdot \mathbb{V}(g_{\mathrm{PP}}^{\lambda}) \leq (1 - \lambda)^2 \sigma^2 + \lambda^2 \left( r\sigma^2 + (1 + r)\sigma_e^2 \right) ,$$

*where $r := n/N$ denotes the ratio of labeled to unlabeled samples.*

From this result, with the proof in Appendix B.1, we can optimally tune $\lambda$ to minimize the variance.

**Corollary 3.3.** *The variance bound in Lemma 3.2 is minimized for*

$$\lambda^* = \frac{1}{1 + r} \cdot \frac{\sigma^2}{\sigma^2 + \sigma_e^2} . \tag{10}$$

*Substituting this value yields the following bound:* $\frac{n}{4} \cdot \mathbb{V}(g_{\mathrm{PP}}^{\lambda^*}) \leq \sigma^2 \cdot \frac{\frac{\sigma_e^2}{\sigma_e^2 + \sigma^2} + r}{1 + r}$ .

Let $V^* := \frac{4\sigma^2}{n} \cdot \frac{(1 + \sigma^2/\sigma_e^2)^{-1} + r}{1 + r}$ denote the optimal variance bound stated in Corollary 3.3. When the pseudo-labels are accurate (namely, $\sigma_e^2 \ll \sigma^2$), $V^*$ approaches $\frac{\sigma^2}{n} \cdot \frac{4r}{1+r}$, reflecting the benefit of incorporating high-quality unlabeled data, which results in substantial variance reduction (as $r \ll 1$). On the other hand, when the pseudo-labels are highly unreliable ($\sigma_e^2 \gg \sigma^2$), the bound approaches $4\sigma^2/n$, effectively recovering the standard (labeled-only) gradient variance (up to a factor of 4, which is asymptotically equivalent); in this case, the optimal interpolation parameter $\lambda^*$ naturally down-weights the gradients from unlabeled data. This observation differs from conventional pseudo-labeling methods, which may reduce empirical variance but typically **introduce bias** when the pseudo-labels are incorrect. In contrast, PP-SSL provides a principled mechanism for leveraging unlabeled data that improves statistical efficiency without compromising correctness. Additionally, in Appendix C, we present a detailed comparison between the optimal parameter $\lambda^*$ derived in Corollary 3.3 and the one proposed in PPI++ [24], within the context of linear regression.

**Convergence rate.** The unbiasedness property (Equation (7)), together with the variance bound (Corollary 3.3), directly implies that using the prediction-powered gradient estimates within first-order methods such as SGD yields a convergence rate of $\mathcal{O}(\sqrt{V^*/T} + 1/T)$ for smooth non-convex functions [29]. Here, $V^*$ can be substantially smaller than the variance of standard gradients estimates, as discussed above. Thus, the reduced variance directly translates to faster convergence.

## 3.2 Prediction-Powered SSL with online tuning

As we have shown in the previous section, proper tuning of $\lambda$ can significantly reduce variance and, consequently, accelerate convergence. Unfortunately, the optimal choice $\lambda^*$ depends on $\sigma^2$ and $\sigma_e^2$ (Equation (10)), which are typically unknown in practice. To address this, we propose an adaptive approach that dynamically adjusts $\lambda$ throughout the optimization process, while achieving performance similar to $\lambda^*$. Specifically, we employ the AdaGrad algorithm [30] to update $\lambda$ online, alongside $w$, without requiring the knowledge of $\sigma^2$ or $\sigma_e^2$. We elaborate on this approach next.

**Online learning and AdaGrad.** Online learning is a decision-making framework in which an algorithm makes predictions and updates them based on information revealed over time. It is particularly well-suited for dynamically evolving environments. Online learning can be described as a sequential game, where in each round $t$, a learner makes a decision $u_t \in \mathcal{D}$, after which the environment reveals a (possibly arbitrary or even adversarially chosen) loss function $h_t$, and the learner incurs a loss of $h_t(u_t)$. The most common metric for evaluating the performance of an online learner is the *regret*, defined as the cumulative difference between the losses incurred by the learner's decisions and those of the best *fixed* decision in-hindsight,

$$\mathcal{R}_T^h := \sum_{t=1}^{T} h_t(u_t) - \min_{u \in \mathcal{D}} \sum_{t=1}^{T} h_t(u) .$$

---

**Algorithm 1:** PP-SSL with Online Tuning

---

1: **Input:** Prediction model $f : \mathcal{X} \to \mathcal{Y}$, learning rate factor $\eta_0$.
2: Initialize $w_1 \in \mathbb{R}^d$, $\lambda_1 \in (0, 1]$
3: **for** $t = 1, \ldots, T$ **do**
4:      Receive $n$ labeled samples $\{(x_t^i, y_t^i)\}_{i=1}^n$ and $N$ unlabeled samples $\{\tilde{x}_t^i\}_{i=1}^N$
5:      Compute stochastic gradients:

$$g_t \leftarrow \frac{1}{n} \sum_{i=1}^n \nabla \ell(w_t; x_t^i, y_t^i), \quad d_t^f \leftarrow \frac{1}{N} \sum_{i=1}^N \nabla \ell(w_t; \tilde{x}_t^i, f(\tilde{x}_t^i)) - \frac{1}{n} \sum_{i=1}^n \nabla \ell(w_t; x_t^i, f(x_t^i))$$

6:      Construct prediction-powered gradient estimate: $g_t^{\lambda_t} \leftarrow g_t + \lambda_t \cdot d_t^f$
7:      Update model parameter: $w_{t+1} \leftarrow w_t - \eta_t g_t^{\lambda_t}$,   where $\eta_t = \eta_0 \left( \sum_{s=1}^t \|g_s^{\lambda_s}\|^2 \right)^{-1/2}$
8:      Define $h_t(\lambda) := \|g_t + \lambda d_t^f\|^2$ and update $\lambda_{t+1}$ using AdaGrad (Equation (12))
9: **end for**

---

One of the most powerful online learning methods is AdaGrad [30], which updates $u_t$ as follows:[2]

$$u_{t+1} = \Pi_{\mathcal{D}}(u_t - \gamma_t \nabla h_t(u_t)), \quad \text{and} \quad \gamma_t := R_{\mathcal{D}} \left( 2 \sum_{s=1}^t \|\nabla h_s(u_s)\|^2 \right)^{-1/2}, \tag{11}$$

where $R_{\mathcal{D}} := \max_{u,v \in \mathcal{D}} \|u - v\|$ is the diameter of $\mathcal{D}$, and $\Pi_{\mathcal{D}}(v) := \arg\min_{u \in \mathcal{D}} \|u - v\|^2$ is the orthogonal projection of $v$ onto $\mathcal{D}$. AdaGrad enjoys the following guarantees; see, e.g., [31].

**Lemma 3.4** (AdaGrad's Regret Bound). *Let $h_1, \ldots, h_T$ be any sequence of convex functions defined over a bounded, convex set $\mathcal{D}$ with diameter $R_{\mathcal{D}}$. Then, for any comparator $u^* \in \mathcal{D}$ and any initial solution $u_1 \in \mathcal{D}$, AdaGrad* (11) *ensures,*

$$\mathcal{R}_T^h(u^*) := \sum_{t=1}^T (h_t(u_t) - h_t(u^*)) \le R_{\mathcal{D}} \sqrt{2 \sum_{t=1}^T \|\nabla h_t(u_t)\|^2}.$$

Importantly, AdaGrad only requires knowledge of the diameter, and otherwise automatically tunes its learning rate $\gamma_t$. In addition, note that AdaGrad's regret bound depends on the sum of squared gradient norms, which will be crucial for the theoretical guarantees we derive.

**Using AdaGrad to dynamically tune $\lambda$.** Recall that our choice of $\lambda^*$ aims to minimize the variance of the gradient estimates. Importantly, the performance of first-order methods like SGD as well as Adam [33] and AdaGrad is directly related to the *cumulative variance of gradient estimates*, i.e., to $\sum_{t=1}^T \mathbb{V}(g_t)$, where $g_t$ is the gradient estimate used at time $t$. Since our prediction-powered gradient estimates employ a parameter $\lambda$, we can denote it as $g_t^\lambda$. This suggests that we can cast the cumulative variance minimization problem as an online learning problem with $h_t : [0, 1] \mapsto \mathbb{R}_+$ defined as,

$$h_t(\lambda) := \|g_t^\lambda\|^2 = \left\| g_t^n + \lambda(\tilde{g}_t^{N,f} - g_t^{n,f}) \right\|^2,$$

where we have used Equation (6) for $g_t^\lambda$. Importantly, it is immediately apparent that $h_t(\cdot)$ is convex in $\lambda$; this enables the use of AdaGrad to dynamically tune $\lambda_t$ in order to reduce the cumulative second moment. Since $g_t^\lambda$ is always (conditionally) unbiased, then minimizing the cumulative second moment is equivalent to minimizing the cumulative variance.

Specifically, for our one-dimensional parameter $\lambda \in [0, 1]$, the AdaGrad update rule is given by:

$$\lambda_{t+1} = \text{clamp}\left(\lambda_t - \gamma_t \nabla h_t(\lambda_t); 0, 1\right), \quad \gamma_t := \left( 2 \sum_{s=1}^t \|\nabla h_s(\lambda_s)\|^2 \right)^{-1/2}, \tag{12}$$

where $\text{clamp}(\cdot; a, b) := \min(\max(a, \cdot), b)$ denotes projection onto the interval $[a, b]$.

---

[2]This is in fact the scalar version of AdaGrad, also known as AdaGrad-Norm [31, 32].

**Overall dynamic approach.** Our dynamic PPI-inspired training approach is depicted in Algorithm 1. At each round, we employ $n$ labeled samples, $N$ unlabeled samples, and the teacher model $f$ to yield a prediction-powered gradient $g_t^{\lambda_t}$, as in Equation (6). The latter is then used to update the model weights $w_{t+1}$ as well as to craft a second moment estimate $h_t(\lambda)$. Then, we update $\lambda_t$ using the AdaGrad update rule on $h_t(\cdot)$, as given in Equation (12). Note that our update rule for $w_t$ also employs an AdaGrad stepsize of the form $\eta_t = \eta_0(\sum_{s=1}^t \|g_s^{\lambda_s}\|^2)^{-1/2}$; this is crucial for properly adapting to $\sigma^2$ and $\sigma_e^2$ without prior knowledge of these quantities. While we concretely employ AdaGrad for simultaneously tuning $w$ and $\lambda$, one can alternatively use any stochastic optimization method (e.g., SGD) for training $w$, and any online learning approach for tuning $\lambda$. As Theorem 3.5 below shows, our specific choice of using AdaGrad for both parameters allows us to *implicitly and optimally* adapt to the unknown $\sigma^2$ and $\sigma_e^2$. This provides a substantial advantage over alternative approaches that require hyperparameter tuning to achieve comparable performance.

The next theorem establishes the convergence of `PP-SSL` (Algorithm 1) and shows that it performs similarly to employing the optimal (yet practically unknown) $\lambda^*$. We provide a proof in Appendix B.

**Theorem 3.5.** *Suppose, in addition to the previous assumptions, that $\mathcal{L}(\cdot)$ is $M$-bounded, i.e., $\max_{w\in\mathbb{R}^d} |\mathcal{L}(w)| \leq M$, and that the stochastic gradients are $G$-bounded, $\|\nabla\ell(w;x,y)\| \leq G$ for all $w \in \mathbb{R}^d, x \in \mathcal{X}, y \in \mathcal{Y}$. Then, Algorithm 1 with a learning rate parameter of $\eta_0 = \sqrt{2M/\beta}$ ensures the following convergence rate:*

$$\mathbb{E}\left[\frac{1}{T}\sum_{t=1}^T \|\nabla\mathcal{L}(w_t)\|^2\right] \leq \mathcal{O}\left(\sqrt{\frac{M\beta V^*}{T}} + \frac{M\beta}{T} + \frac{\sqrt{M\beta}G}{T}\right),$$

*where $V^*$ is the variance bound obtained for the optimal choice of $\lambda^*$ (see Corollary 3.3).*

The convergence rate in Theorem 3.5 consists of three terms. The first two terms correspond to the rate we would obtain upon using $\lambda^*$ throughout all updates. However, since we do not know $\lambda^*$ in advance and instead learn it during training, we incur an additional term of order $\sqrt{M\beta}G/T$. Fortunately, this extra term decays substantially faster than the leading term, which decays as $\mathcal{O}(\sqrt{V^*/T})$. Therefore, the overall convergence rate is of order $\mathcal{O}(\sqrt{V^*/T})$, matching the rate with the optimal choice of $\lambda^*$.

## 4 Experiments

In this section, we demonstrate the benefits of our proposed method through a series of experiments. We evaluate performance on both a synthetic dataset (Section 4.1) and three real-world datasets (Sections 4.2, 4.3.1 and 4.3.2). Our experiments encompass both regression and classification tasks. For regression tasks, we report Mean Absolute Error (MAE), Mean Squared Error (MSE), and the coefficient of determination ($R^2$). For classification tasks, we report accuracy. Additional experiments analyzing the choice and dynamics of $\lambda$ are provided in Appendix H.

**Baseline methods.** We compare our proposed `PP-SSL` method to 4 baseline training methods: **(1)** `Teacher` model $f$ used for pseudo-labeling; **(2)** `Only Labeled` model that is trained only on limited labeled data ($n$ samples) by minimizing (1); **(3)** `SSL` model trained both on labeled and pseudo-labeled data ($n + N$ samples) by minimizing (2); and **(4)** `PPI++` model trained both on labeled and pseudo-labeled data ($n+N$ samples) minimizing the debiased SSL loss (4) with $\lambda = \lambda_{\text{PP}}^*$ from [24].[3] Importantly, our `PP-SSL` model is also trained both on the same limited labeled data and pseudo-labeled data, minimizing (4) but with $\lambda$ obtained by Algorithm 1.

### 4.1 Experiments on synthetic data with a linear model

**Dataset.** We design a synthetic linear regression experiment with two groups to simulate a scenario in which the `Teacher` model exhibits different performance across groups. Each input has $m = 10$ features: the first nine are drawn from a standard normal distribution, and the last is a binary indicator denoting group membership. For group A, labels are generated using a linear model with additive zero-mean Gaussian noise. Group A comprises the 80% of samples with the smallest clean labels, i.e., the outputs of the true model before adding noise. The remaining samples form group B, where

---

[3]Vanilla PPI is omitted due to its inferior performance compared to `PPI++` with tunable $\lambda$.

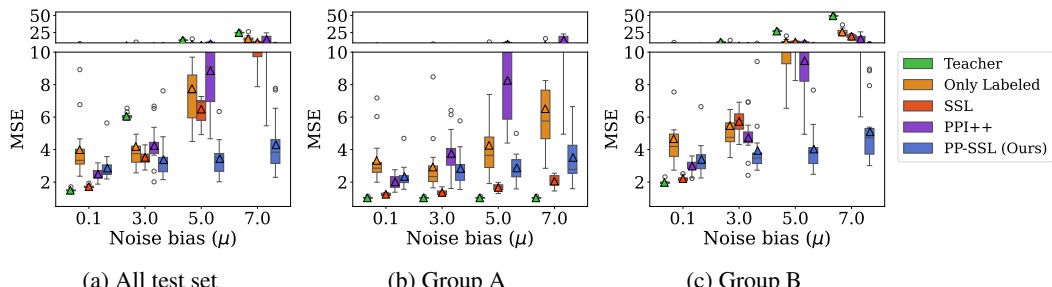

(a) All test set         (b) Group A         (c) Group B

Figure 1: MSE on synthetic data as a function of noise bias $\mu$. Results are shown for: (a) the full test set; (b) Group A, where the teacher model is accurate (oracle-like); and (c) Group B, which includes additive biased noise. Results evaluated on 100 independent experiments.

labels follow the same linear model but include an additional biased Gaussian noise term with mean $\mu$, which controls the magnitude of the bias. This induces a distributional shift between the two groups. The `Teacher` is a fixed linear regressor that serves as an oracle for group A by using the true model weights for prediction.

**Experimental setup.** We set $n = 20$, $N = 1{,}000$, and $n_{\text{test}} = 1{,}000$. We vary the noise bias $\mu \in \{0.1, 1, 3, 5, 7\}$ to control the intensity of the distributional shift. All methods (except the `Teacher`) implemented by fitting a linear regression model using ADAM optimizer with the same hyper-parameters and batch size. See Appendix D for more details on data generation process, training schemes, additional experimental setups, and results that include the $R^2$ metric.

**Main Results.** Figure 1 presents group-wise MSE obtained by different models across varying group B's noise bias levels $\mu$. As can be seen, the average MSE across the two groups (left panel) increases as $\mu$ grows, but our `PP-SSL` shows better performance, demonstrating the advantage of using debiased SSL risk with adaptive $\lambda$ when the pseudo-labels are biased. In group A (middle), the `Teacher` model provides oracle predictions by design, and thus the `SSL` method has strong performance. However, in group B (right) the `SSL` model performs even worse than the `Only Labeled` model as the `Teacher` performs poorly. This is in striking contrast with our `PP-SSL` that tends to achieve the best MSE for that group, especially in the high bias regimes. Crucially, `PP-SSL` outperforms `PPI++` although the two use a debiased risk but with different strategies to tune $\lambda$.

In the above experiment, all models have access to the group indicator during both training and testing. In Appendix D, we report similar results for the setting where the group indicator is not provided to the models.

**Adaptive tuning of $\lambda$.** To evaluate the benefit of adaptively tuning $\lambda$, we conduct a targeted experiment comparing `PP-SSL` with a `PPI++`-inspired baseline, that employs prediction-powered gradients (see Eq. (6)) but uses a fixed $\lambda$ value throughout training. In Figure 2, we show the final MSE on the test set for `PP-SSL` (dashed line) and the baseline for different $\lambda$ values. Our adaptive method consistently achieves performance comparable to the baseline with the best fixed $\lambda$ value. The optimal error depends on the teacher's quality (bias), with smaller $\lambda$ values preferred for highly biased teachers, which aligns with our theory. These results demonstrate that our method automatically adjusts to the teacher's quality, achieving near-optimal performance.

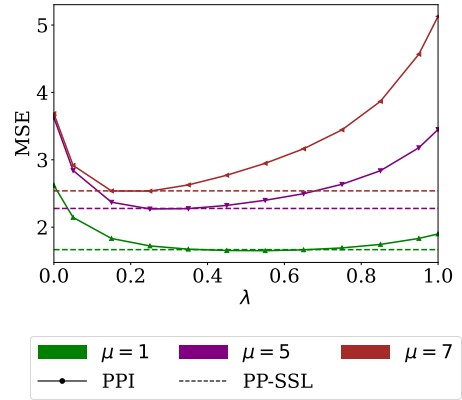

Figure 2: Final test set MSE for `PP-SSL` and PPI baseline with constant $\lambda$ for varying $\lambda$ values.

## 4.2 Experiments on real tabular data with a linear model

**Dataset.** We evaluate our method on the "California Housing" dataset [34], which contains 8 numeric features and a target variable representing house prices, across $20{,}640$ samples. To demonstrate the

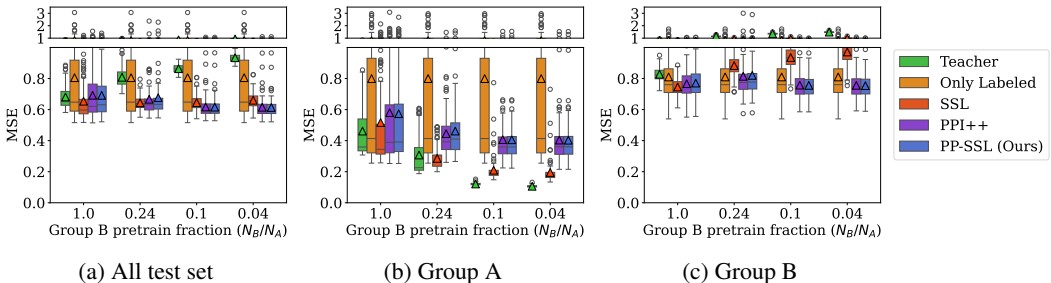

(a) All test set    (b) Group A    (c) Group B

Figure 3: Results for the California Housing dataset: MSE of various methods as a function of $N_B/N_A$—the fraction of samples from each group used to train the `Teacher` model; e.g., $N_B/N_A = 0.5$ means twice as many group A samples as group B. Results correspond to 100 data splits.

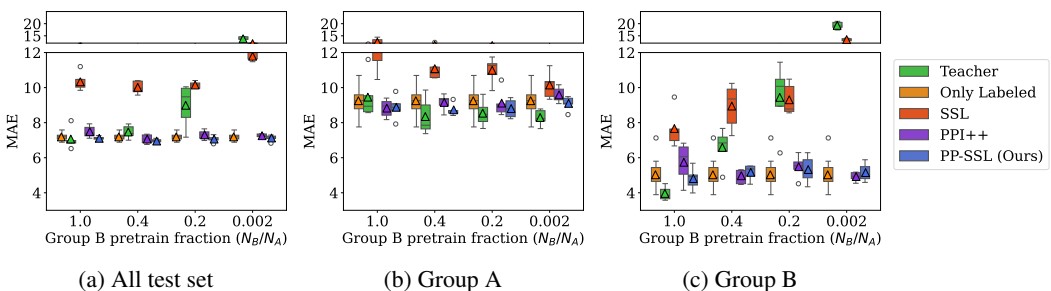

(a) All test set    (b) Group A    (c) Group B

Figure 4: Results for age estimation: MAE as a function of $N_B/N_A$, across 5 data splits.

importance of the debiased SSL risk we split the data into two groups. Group A includes the $(x^i, y^i)$ with the 40% lowest target $y^i$; otherwise the samples are assigned to group B.

**Experimental setup and results.** We split the data by setting $n \approx 100$, $N \approx 18{,}000$, $n_{\text{val}} \approx 1{,}000$ for validation, and $n_{\text{test}} \approx 1{,}000$ for testing. For all methods, we train a linear model. The `Teacher` model is trained on $N_A \approx 50$ disjoint labeled samples from group A, and a varied number of samples from group B, denoted by $N_B$, in the range of 5 to about 50. See Appendix E for more details. Following Figure 3, we can see that as $N_B/N_A$ increases and the `Teacher`'s performance on group B improves, and all methods show reduced MSE values. `PPI++` and `PP-SSL` have comparable MSE across the board, where both methods tend to outperform the other baselines as the teacher becomes less accurate due to the limited data from Group B. Additional experiments in Appendix E further support these observations.

### 4.3 Experiments on real visual data with a deep neural network

#### 4.3.1 UTKFace data

**Dataset.** We use the facial age estimation UTKFace dataset [35]. This dataset contains about 20,000 face images with age annotations ranging from 0 to 116 years. We select the aligned and cropped version of the dataset, and resize the images to $224 \times 224$ pixels.

**Experimental setup and results.** We repeat a similar setup to that from Section 4.2. Specifically, we split the data by setting $n \approx 700$, $N \approx 16{,}500$, $n_{\text{val}} \approx 2{,}000$, and $n_{\text{test}} \approx 2{,}000$. For all methods, we train a ResNet50 model [36]. The teacher model is trained on $N_A \approx 800$ disjoint labeled samples from group A (ages over 30) and a varied number of samples from group B (ages 0-29), with $N_B$ ranging from 10 to about 800. As portrayed in Figure 4, the `SSL` method is highly sensitive to the performance of the `Teacher` model: observe how `SSL` performs worse than the `Only Labeled` approach. By contrast, `PPI++` and `PP-SSL` achieve lower MSE values, where our method has a noticeable advantage. This experiment shows that the debiasing approach can offer performance gains over the `Only Labeled` baseline, even when the `Teacher` model is relatively weak. In Appendix F we provide additional implementation details and results based on MSE and $R^2$ metrics.

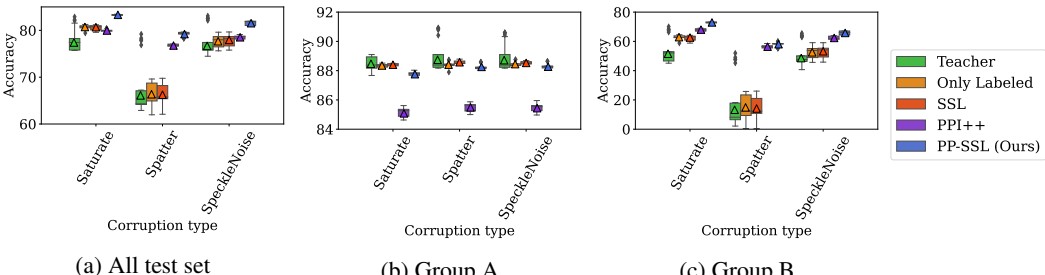

Figure 5: Accuracy on CIFAR-10 data as a function of corruption type. Results are shown for: (a) the full test set; (b) Group A, clean images; and (c) Group B, corrupted images. Results evaluated on 5 different seeds.

### 4.3.2 CIFAR-10 data

**Dataset.** We evaluate our method on the CIFAR-10 dataset [37] and its corrupted variant, CIFAR-10-C [38]. CIFAR-10 contains $60,000$ images of size $32 \times 32$ from 10 classes, split into $50,000$ training images and $10,000$ test images.

**Experimental setup and results.** To simulate a biased teacher, we split the dataset into two groups: Group B consists of the `dog`, `cat`, and `deer` classes from CIFAR-10-C (corrupted samples), while Group A consists of the remaining clean CIFAR-10 classes. Corruptions such as Saturate, Spatter, and Speckle Noise from the CIFAR-10-C benchmark were applied to both training and test samples in Group B. We followed a standard SSL protocol, where features are extracted using a ResNet50 pretrained on ImageNet, and a linear classifier was trained on top. The teacher model is a linear classifier trained on a small clean subset. See Appendix G for additional details. As shown in Figure 5, `PP-SSL` consistently outperforms all baselines in terms of both total accuracy and Group B (corrupted) accuracy, while maintaining comparable performance on Group A (clean). The performance gap on Group B varies with the corruption type, reflecting the differing degrees of pseudo-label inaccuracy under different distortions. Additional corruption types and severity levels yielded similar trends, which we omit here for brevity.

## 5 Discussion

We introduce `PP-SSL`, a semi-supervised online learning framework that incorporates a debiased empirical risk to account for errors in pseudo-labels generated by a teacher model. We demonstrate that our approach to dynamically update the interpolation parameter, which governs the reliance on pseudo-labeled data during training, consistently enhances performance compared to baseline methods, especially when the teacher model exhibits degraded performance on a specific subgroup.

A key limitation of our method, shared by many SSL methods, is the assumption that labeled and unlabeled data are drawn from the same underlying distribution. In practice, unlabeled examples may come from different domains or be produced by generative models, introducing distributional shifts and bias. Extending our framework to handle such shifts is a natural direction for future work. For instance, importance weighting can be introduced to correct for covariate shift, either by reweighting the gradient estimator used for $\lambda$ adaptation or by modifying the loss itself. More complex shifts, such as those induced by synthetic or generative data, would likely require adaptations to both the pseudo-labeling mechanism and the gradient-based weighting strategy.

Our theoretical analysis provides the first gradient variance bounds for PPI-based methods, yet assumes a fixed teacher for tractability. Extending these guarantees to self-training scenarios with evolving teachers remains challenging due to non-stationarity and requires more nuanced dynamic regret analysis. While not currently covered by our theory, `PP-SSL` remains practically applicable to self-training, where the model iteratively generates pseudo-labels the PPI loss remains unbiased.

Overall, our work offers a principled step toward more adaptive and reliable SSL, with both theoretical and practical implications. Similar to works in this field, this research has potential social implications.

## Acknowledgments and Disclosure of Funding

NS, SS, and YR were supported by the European Union (ERC, SafetyBounds, 101163414). Views and opinions expressed are however those of the authors only and do not necessarily reflect those of the European Union or the European Research Council Executive Agency. Neither the European Union nor the granting authority can be held responsible for them. NS, SS, and YR were also partially supported by the Israel Science Foundation (ISF grant 729/21). YR acknowledges additional support from the Career Advancement Fellowship at the Technion. KYL and RD were partially supported by Israel PBC-VATAT, by the Technion Artificial Intelligence Hub (Tech.AI), and by the Israel Science Foundation (grant No. 3109/24).

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

# A Relating the teacher's prediction error to the gradient variance

In this section, we prove Lemma 3.1, which establishes a connection between the teacher model's prediction error $\mathcal{E}^f = \mathbb{E}_{x,y \sim P}(y - f(x))^2$ and the variance bound of the error loss gradient $\sigma_e^2$, defined in Equation (9). This result holds for any loss function whose gradient $\nabla \ell(w; x, y)$ is $L_Y$-Lipschitz in $y$. We then show that both the squared loss and the logistic loss satisfy this condition.

For ease of reference, we state Lemma 3.1 from the main manuscript.

**Lemma 3.1.** Assume that the gradient $\nabla \ell(w; x, y)$ is $L_Y$-Lipschitz in $y$, i.e., for any $w \in \mathbb{R}^d$, $x \in \mathcal{X}$, and $y_1, y_2 \in \mathcal{Y}$, we have $\|\nabla \ell(w; x, y_1) - \nabla \ell(w; x, y_2)\| \leq L_Y |y_1 - y_2|$. Then, $\sigma_e^2 \leq L_Y^2 \cdot \mathcal{E}^f$.
*Proof of Lemma 3.1.* Recall that $\ell_e^f(w; x, y) = \ell(w; x, y) - \ell(w; x, f(x))$. Since $\nabla \ell(w; x, y)$ is

$L_Y$-Lipschitz in $y$, we have:

$$\mathbb{E} \left\| \nabla \ell_e^f(w; x, y) \right\|^2 = \mathbb{E} \left\| \nabla \ell(w; x, y) - \nabla \ell(w; x, f(x)) \right\|^2 \leq \mathbb{E}[L_Y^2 (y - f(x))^2] = L_Y^2 \mathcal{E}^f .$$

Since the variance is always upper bounded by the second moment, we get:

$$\mathbb{E} \left\| \nabla \ell_e^f(w; x, y) - \nabla \mathcal{L}_e^f(w) \right\|^2 \leq \mathbb{E} \left\| \nabla \ell_e^f(w; x, y) \right\|^2 \leq L_Y^2 \mathcal{E}^f .$$

Finally, as this bound holds for any $w \in \mathbb{R}^d$, it also holds for the supremum, and thus $\sigma_e^2 \leq L_Y^2 \mathcal{E}^f$. $\square$

Next, we consider the squared and logistic losses and show that their gradients are Lipschitz with respect to the $y$ input. Let $\phi_w : \mathcal{X} \to \mathcal{Y}$ denote a model parameterized by weights $w \in \mathbb{R}^d$, and assume that $\|\nabla_w \phi_w(x)\| \leq R$ for all $w \in \mathbb{R}^d$ and $x \in \mathcal{X}$. For instance, for a linear model $\phi_w(x) = w^\top x$, this condition holds when the input features are bounded, i.e., $\|x\| \leq R$.

**Squared loss.** The squared $L_2$ loss is defined as $\ell_{L_2}(w; x, y) = \frac{1}{2} (\phi_w(x) - y)^2$ with gradient $\nabla \ell_{L_2}(w; x, y) = (\phi_w(x) - y) \nabla_w \phi_w(x)$. Therefore, for any $y_1, y_2 \in \mathcal{Y}$, we have:

$$\|\nabla \ell_{L_2}(w; x, y_1) - \nabla \ell_{L_2}(w; x, y_2)\| = \|(y_2 - y_1) \nabla_w \phi_w(x)\| = |y_1 - y_2| \|\nabla_w \phi_w(x)\|$$
$$\leq R \cdot |y_1 - y_2| .$$

Therefore, the squared loss gradient is Lipschitz with coefficient $R$.

**Logistic loss.** The logistic loss is defined as $\ell_{\log.}(w; x, y) = -y \log (\sigma(\phi_w(x))) - (1 - y) \log (1 - \sigma(\phi_w(x)))$, where $\sigma(u) = 1/(1 + e^{-x})$ is the sigmoid function. Its gradient is given by $\nabla \ell_{\log.}(w; x, y) = (\sigma(\phi_w(x)) - y) \nabla_w \phi_w(x)$. Thus, for any $y_1, y_2 \in \mathcal{Y}$, we have:

$$\|\nabla \ell_{\log.}(w; x, y_1) - \nabla \ell_{\log.}(w; x, y_2)\| = \|(y_2 - y_1) \nabla_w \phi_w(x)\| = |y_1 - y_2| \|\nabla_w \phi_w(x)\|$$
$$\leq R \cdot |y_1 - y_2| .$$

Similarly to the squared loss, the logistic loss gradient is also Lipschitz with coefficient $R$. For both cases, Lemma 3.1 implies that $\sigma_e^2$ is bounded by $R^2 \mathcal{E}^f$.

# B Theoretical analysis of prediction-powered gradients and `PP-SSL`

In this section, we begin by establishing the variance bound of the prediction-powered gradient in Appendix B.1. Then, in Appendix B.2, we prove the convergence result for our `PP-SSL` method, as stated in Theorem 3.5.

## B.1 Proof of Lemma 3.2

*Proof.* Let us define $\tilde{g}^N := \frac{1}{N} \sum_{i=1}^{N} \nabla \ell(w; \tilde{x}^i, \tilde{y}^i)$, where $\tilde{y}^i$ is the *true* (unknown) label of $\tilde{x}^i$, i.e., $\tilde{y}^i \sim \mathbb{P}_{Y|X}(\cdot | \tilde{x}^i)$. Thus, we can decompose $g_{\text{PP}}^1 := g^n + \tilde{g}^{N,f} - g^{n,f}$ as follows:

$$g_{\text{PP}}^1 = \tilde{g}^N + \left( \tilde{g}^{N,f} - \tilde{g}^N \right) - \left( \nabla \mathcal{L}^f(w) - \nabla \mathcal{L}(w) \right) + \left( g^n - g^{n,f} \right) - \left( \nabla \mathcal{L}(w) - \nabla \mathcal{L}^f(w) \right) ,$$

where $\mathcal{L}^f(w) := \mathbb{E}_{x \sim \mathbb{P}_X}[\ell(w; x, f(x))]$. Note that $g_{\mathrm{PP}}^\lambda = (1 - \lambda)g^n + \lambda g_{\mathrm{PP}}^1$. Considering the variance of $g_{\mathrm{PP}}^\lambda$ and substituting this expression, we obtain:

$$
\begin{aligned}
\mathbb{V}(g_{\mathrm{PP}}^\lambda) &= \mathbb{E}\|g_{\mathrm{PP}}^\lambda - \nabla\mathcal{L}(w)\|^2 \\
&= \mathbb{E}\left\|(1 - \lambda)\left(g^n - \nabla\mathcal{L}(w)\right) + \lambda\left(g_{\mathrm{PP}}^1 - \nabla\mathcal{L}(w)\right)\right\|^2 \\
&\leq 4(1 - \lambda)^2 \mathbb{E}\left\|g^n - \nabla\mathcal{L}(w)\right\|^2 + 4\lambda^2 \mathbb{E}\left\|\tilde{g}^N - \nabla\mathcal{L}(w)\right\|^2 + \\
&\quad 4\lambda^2 \mathbb{E}\left\|\left(\tilde{g}^N - \tilde{g}^{N,f}\right) - \nabla\mathcal{L}_e^f(w)\right\|^2 + 4\lambda^2 \mathbb{E}\left\|\left(g^n - g^{n,f}\right) - \nabla\mathcal{L}_e^f(w)\right\|^2 \\
&= 4(1 - \lambda)^2 \mathbb{V}(g^n) + 4\lambda^2\left(\mathbb{V}(\tilde{g}^N) + \mathbb{V}(\tilde{g}^N - \tilde{g}^{N,f}) + \mathbb{V}(g^n - g^{n,f})\right),
\end{aligned}
$$

where the first inequality follows from $\|\sum_{i=1}^n u_i\|^2 \leq n \sum_{i=1}^n \|u_i\|^2$, and we also used $\mathcal{L}_e^f(w) = \mathcal{L}(w) - \mathcal{L}^f(w)$. Dividing by 4 and plugging the variance bounds from Equations (8) and (9) (accounting for mini-batch gradients) yields:

$$
\frac{\mathbb{V}(g_{\mathrm{PP}}^\lambda)}{4} \leq (1 - \lambda)^2 \cdot \frac{\sigma^2}{n} + \lambda^2\left(\frac{\sigma^2}{N} + \frac{\sigma_e^2}{N} + \frac{\sigma_e^2}{n}\right) = \frac{(1 - \lambda)^2 \sigma^2 + \lambda^2\left(r\sigma^2 + (1 + r)\sigma_e^2\right)}{n},
$$

where $1/N = r/n$. Finally, multiplying by $n$ concludes the proof. $\qquad\square$

## B.2  Proof of Theorem 3.5

*Proof.* Recall that we use the AdaGrad stepsize to update $w_t$, i.e., for some parameter $\eta_0 > 0$, we update $w_t$ as follows:

$$
w_{t+1} = w_t - \eta_t g_t^{\lambda_t}, \quad \text{where} \quad \eta_t = \frac{\eta_0}{\sqrt{\sum_{s=1}^t \|g_s^{\lambda_s}\|^2}}. \tag{AdaGrad-Norm}
$$

Therefore, we can employ the following lemma (see, e.g., Lemma G.3 in [39])

**Lemma B.1.** *For a $\beta$-smooth and $M$-bounded function (i.e., $\max_{w \in \mathbb{R}^d} |\mathcal{L}(w)| \leq M$), the iterates $w_1, \ldots, w_T$ defined by the above* (AdaGrad-Norm) *update rule satisfy:*

$$
\sum_{t=1}^T \|\nabla\mathcal{L}(w_t)\|^2 \leq \left(\frac{2M}{\eta_0} + \eta_0\beta\right)\sqrt{\sum_{t=1}^T \|g_t^{\lambda_t}\|^2} + \sum_{t=1}^T (\nabla\mathcal{L}(w_t) - g_t^{\lambda_t})^\top \nabla\mathcal{L}(w_t).
$$

Now, recalling that $g_t^{\lambda_t}$ is an unbiased estimator of $\nabla\mathcal{L}(w_t)$, in the sense that $\mathbb{E}_{t-1}[g_t^{\lambda_t}] = \nabla\mathcal{L}(w_t)$, taking expectation of the bound above yields:

$$
\begin{aligned}
\sum_{t=1}^T \mathbb{E}\|\nabla\mathcal{L}(w_t)\|^2 &\leq \left(\frac{2M}{\eta_0} + \eta_0\beta\right)\mathbb{E}\sqrt{\sum_{t=1}^T \|g_t^{\lambda_t}\|^2} \\
&= \left(\frac{2M}{\eta_0} + \eta_0\beta\right)\mathbb{E}\sqrt{\sum_{t=1}^T h_t(\lambda_t)} \\
&\leq \left(\frac{2M}{\eta_0} + \eta_0\beta\right)\mathbb{E}\sqrt{\sum_{t=1}^T \mathcal{R}_T^h(\lambda^*) + \sum_{t=1}^T h_t(\lambda^*)}, \tag{13}
\end{aligned}
$$

where the equality follows from the definition $h_t(\lambda) := \|g_t^\lambda\|^2$, and the last inequality uses the definition of regret, $\mathcal{R}_T^h(\lambda^*) := \sum_{t=1}^T h_t(\lambda_t) - \sum_{t=1}^T h_t(\lambda^*)$.

Since we update $\lambda_t$ using AdaGrad as well (see Equation (12)), we can bound the regret $\mathcal{R}_T^h(\lambda^*)$, as formalized in the following lemma (proved in Appendix B.3).

**Lemma B.2.** *Assume the stochastic gradients are $G$-bounded; that is, for any $w \in \mathbb{R}^d$ and $(x, y) \in \mathcal{X} \times \mathcal{Y}$, we have $\|\nabla\ell(w; x, y)\| \leq G$. Then, the 1D-AdaGrad algorithm used in Equation (12) yields the following regret bound:*

$$
\mathcal{R}_T^h(\lambda^*) \leq 128G^2 + 8G\sqrt{2\sum_{t=1}^T h_t(\lambda^*)}.
$$

Plugging this regret bound into Equation (13) gives:

$$\sum_{t=1}^{T} \mathbb{E}\|\nabla\mathcal{L}(w_t)\|^2 \leq \left(\frac{2M}{\eta_0} + \eta_0\beta\right)\mathbb{E}\sqrt{128G^2 + 8G\sqrt{2\sum_{t=1}^{T} h_t(\lambda^*)} + \sum_{t=1}^{T} h_t(\lambda^*)} . \qquad (14)$$

Observe that the last two terms under the square root take the form $8G\sqrt{2A} + A$, where $A = \sum_{t=1}^{T} h_t(\lambda^*)$, which we can bound using the helper Lemma B.3 below, which we prove in Appendix B.4.

**Lemma B.3.** *For any $A, C > 0$, it holds that $C\sqrt{A} + A \leq 2C^2 + 2A$.*

Hence, from Equation (14), it follows that:

$$\sum_{t=1}^{T} \mathbb{E}\|\nabla\mathcal{L}(w_t)\|^2 \leq \left(\frac{2M}{\eta_0} + \eta_0\beta\right)\mathbb{E}\sqrt{128G^2 + 256G^2 + 2\sum_{t=1}^{T} h_t(\lambda^*)}$$

$$\leq \left(\frac{2M}{\eta_0} + \eta_0\beta\right)8\sqrt{6}G + \left(\frac{2M}{\eta_0} + \eta_0\beta\right)\mathbb{E}\sqrt{2\sum_{t=1}^{T} h_t(\lambda^*)}$$

$$\leq \left(\frac{2M}{\eta_0} + \eta_0\beta\right)8\sqrt{6}G + \left(\frac{2M}{\eta_0} + \eta_0\beta\right)\sqrt{2\sum_{t=1}^{T} \mathbb{E}h_t(\lambda^*)} , \qquad (15)$$

where the second inequality holds because $\sqrt{a+b} \leq \sqrt{a} + \sqrt{b}$ for any $a, b \geq 0$, and the last inequality follows from Jensen's inequality applied to the concave function $H(z) := \sqrt{z}$. Next, focusing on $\mathbb{E}h_t(\lambda^*)$, we apply the law of total expectation to obtain:

$$\mathbb{E}h_t(\lambda^*) = \mathbb{E}\|g_t^{\lambda^*}\|^2 = \mathbb{E}\mathbb{E}_{t-1}\|g_t^{\lambda^*}\|^2 \overset{(\dagger)}{=} \mathbb{E}[\mathbb{V}_{t-1}(g_t^{\lambda^*}) + \|\mathbb{E}_{t-1}g_t^{\lambda^*}\|^2]$$

$$\overset{(\ddagger)}{=} \mathbb{E}[\mathbb{V}_{t-1}(g_t^{\lambda^*})] + \mathbb{E}\|\nabla\mathcal{L}(w_t)\|^2 .$$

where $(\dagger)$ follows from the definition of conditional variance and $(\ddagger)$ is due to the (conditional) unbiasedness of $g_t^{\lambda^*}$. Note that the conditional variance with the optimal $\lambda^*$, $\mathbb{V}_{t-1}(g_t^{\lambda^*})$, is bounded by $V^* := 4\kappa\sigma^2/n$ according to Corollary 3.3, where $\kappa := \frac{(1+\sigma^2/\sigma_e^2)^{-1}+r}{1+r}$. Therefore,

$$\mathbb{E}h_t(\lambda^*) \leq V^* + \mathbb{E}\|\nabla\mathcal{L}(w_t)\|^2 .$$

Substituting this back into Equation (15), we get:

$$\sum_{t=1}^{T} \mathbb{E}\|\nabla\mathcal{L}(w_t)\|^2 \leq \left(\frac{2M}{\eta_0} + \eta_0\beta\right)8\sqrt{6}G + \left(\frac{2M}{\eta_0} + \eta_0\beta\right)\sqrt{2V^*T + 2\sum_{t=1}^{T} \mathbb{E}\|\nabla\mathcal{L}(w_t)\|^2}$$

$$\leq \left(\frac{2M}{\eta_0} + \eta_0\beta\right)8\sqrt{6}G + \left(\frac{2M}{\eta_0} + \eta_0\beta\right)\left(\sqrt{2V^*T} + \sqrt{2\sum_{t=1}^{T} \mathbb{E}\|\nabla\mathcal{L}(w_t)\|^2}\right) .$$
$$(16)$$

We can now solve this inequality for $\sum_{t=1}^{T} \mathbb{E}\|\nabla\mathcal{L}(w_t)\|^2$ using the following lemma, which we prove in Appendix B.5.

**Lemma B.4.** *Let $A, B, C > 0$. If $A \leq B\sqrt{A} + C$, then $A \leq 4B^2 + 2C$.*

Applying Lemma B.4 to the inequality in Equation (16), we get:

$$\sum_{t=1}^{T} \mathbb{E}\|\nabla\mathcal{L}(w_t)\|^2 \leq 2\left(\frac{2M}{\eta_0} + \eta_0\beta\right)\left(8\sqrt{6}G + \sqrt{2V^*T}\right) + 8\left(\frac{2M}{\eta_0} + \eta_0\beta\right)^2 .$$

Setting $\eta_0 = \sqrt{2M/\beta}$ gives the following bound:

$$\sum_{t=1}^{T} \mathbb{E}\|\nabla \mathcal{L}(w_t)\|^2 \leq 64\sqrt{3M\beta}G + 8\sqrt{M\beta V^* T} + 64M\beta .$$

Finally, dividing by $T$ concludes the proof. □

## B.3   Proof of Lemma B.2

*Proof.* By the definition of $h_t(\lambda) = \|g_t^\lambda\|^2 = \|g_t + \lambda d_t^f\|^2$, the gradient (derivative) of $h_t$ is given by $\nabla h_t(\lambda) = 2\langle d_t^f, g_t^\lambda \rangle$. Applying the Cauchy-Schwarz inequality, we can bound this gradient norm as follows:

$$\|\nabla h_t(\lambda_t)\|^2 \leq 4\|d_t^f\|^2\|g_t^{\lambda_t}\|^2 = 4\|d_t^f\|^2 h_t(\lambda_t) \leq 16G^2 h_t(\lambda_t) , \tag{17}$$

where the last inequality follows from the bounded gradients assumption and the triangle inequality:

$$\|d_t^f\| = \left\| \frac{1}{N}\sum_{i=1}^{N} \nabla \ell(w_t; \tilde{x}_t^i, f(\tilde{x}_t^i)) - \frac{1}{n}\sum_{i=1}^{n} \nabla \ell(w_t; x_t^i, f(x_t^i)) \right\|$$

$$\leq \frac{1}{N}\sum_{i=1}^{N} \left\| \nabla \ell(w_t; \tilde{x}_t^i, f(\tilde{x}_t^i)) \right\| + \frac{1}{n}\sum_{i=1}^{n} \left\| \nabla \ell(w_t; x_t^i, f(x_t^i)) \right\| \leq 2G .$$

Substituting Equation (17) into the AdaGrad regret bound from Lemma 3.4, and noting that for our case $\mathcal{D} = [0, 1]$ with diameter $R_{\mathcal{D}} = 1$, we obtain:

$$\mathcal{R}_T^h(\lambda^*) \leq \sqrt{2\sum_{t=1}^{T} \|\nabla h_t(\lambda_t)\|^2} \leq \sqrt{32G^2 \sum_{t=1}^{T} h_t(\lambda_t)}$$

$$= 4G\sqrt{2\mathcal{R}_T^h(\lambda^*) + 2\sum_{t=1}^{T} h_t(\lambda^*)}$$

$$\leq 4G\sqrt{2\mathcal{R}_T^h(\lambda^*)} + 4G\sqrt{2\sum_{t=1}^{T} h_t(\lambda^*)} ,$$

where the last inequality is due to $\sqrt{a+b} \leq \sqrt{a} + \sqrt{b}$. Solving this inequality for $\mathcal{R}_T^h(\lambda^*)$ (using Lemma B.4) yields the result:

$$\mathcal{R}_T^h(\lambda^*) \leq 128G^2 + 8G\sqrt{2\sum_{t=1}^{T} h_t(\lambda^*)} .$$

□

## B.4   Proof of Lemma B.3

*Proof.* We consider two cases:

**Case 1: $A \geq C\sqrt{A}$.**   In this case, $C\sqrt{A} + A \leq 2A$.

**Case 2: $A \leq C\sqrt{A}$.**   Dividing both sides by $\sqrt{A}$, we obtain $\sqrt{A} \leq C$. Hence:

$$C\sqrt{A} + A \leq 2C\sqrt{A} \leq 2C^2 .$$

Combining both cases, we have $C\sqrt{A} + A \leq \max\{2A, 2C^2\} \leq 2C^2 + 2A$, which concludes the proof. □

## B.5   Proof of Lemma B.4

*Proof.* Consider two cases. First, if $B\sqrt{A} \geq C$, then $A \leq 2B\sqrt{A}$, which implies $A \leq 4B^2$. Second, if $B\sqrt{A} \leq C$, then $A \leq 2C$. In both cases, we conclude that $A \leq \max\{4B^2, 2C\} \leq 4B^2 + 2C$. □

## C  A comparative analysis of optimal $\lambda$ tuning for `PPI++` and `PP-SSL`

Next, we theoretically compare the optimal weighting parameter $\lambda$ in our method (`PP-SSL`) and in Prediction-Powered Inference (`PPI++`). This parameter plays a crucial role in balancing the influence of labeled and unlabeled data in both approaches, directly affecting the variance. While both methods aim to reduce variance in parameter estimation, they derive the optimal $\lambda$ from different optimization objectives. We analyze the derivations of the original PPI++ expression, and our variance bound minimization from Corollary 3.3. For clarity and concrete interpretation, we focus on linear regression with mean squared error loss throughout our analysis.

### C.1  PPI++ expression

In this section, we analyze the optimal $\lambda$ expression defined in [24, Proposition 2]:

$$\lambda_{PP}^* = \frac{\text{Tr}(H_{w^*}^{-1}(\text{Cov}(\nabla\ell_{w^*},\nabla\ell_{w^*}^f) + \text{Cov}(\nabla\ell_{w^*}^f,\nabla\ell_{w^*}))H_{w^*}^{-1})}{2(1+r)\text{Tr}(H_{w^*}^{-1}\text{Cov}(\nabla\ell_{w^*}^f)H_{w^*}^{-1})},$$

where $H_{w^*}$ is the Hessian of loss ($\nabla^2\ell_{w^*}$) with the optimal parameter $w^*$.

For linear regression with MSE loss, we consider a model $\hat{y} = x^\top w$ with loss $\ell(w,(x,y)) = \frac{1}{2}(y - x^\top w)^2$ and prediction loss $\ell^f(w,(x,f(x))) = \frac{1}{2}(f(x) - x^\top w)^2$. This yields the following gradients:

$$\nabla\ell_w = x(x^\top w - y), \quad \nabla\ell_w^f = x(x^\top w - f(x)),$$

with Hessian $H_w = xx^T$, which at $w^*$ becomes $H_{w^*} = \mathbb{E}[xx^T] = \Sigma_x$.

We define the true residual $\varepsilon = y - x^\top w^*$ and pseudo-label residual $\varepsilon_f = f(x) - x^\top w^*$, allowing us to express the gradients as $\nabla\ell_{w^*} = -x\varepsilon$ and $\nabla\ell_{w^*}^f = -x\varepsilon_f$. Since we use the true regression coefficient vector $w^*$, it is reasonable to assume independence between features $x$ and residuals $\varepsilon, \varepsilon_f$. Under this simplified assumption, the covariance terms become $\text{Cov}(\nabla\ell_{w^*},\nabla\ell_{w^*}^f) = \mathbb{E}[xx^T\varepsilon\varepsilon_f] := \Sigma_x \cdot \text{Cov}(\varepsilon,\varepsilon_f)$ and $\text{Cov}(\nabla\ell_{w^*}^f) = \mathbb{E}[xx^T\varepsilon_f^2] := \Sigma_x \cdot \text{Var}(\varepsilon_f)$.

We now substitute into the optimal $\lambda$ formula:

$$\lambda^* = \frac{\text{Tr}(\Sigma_x^{-1}(\Sigma_x \cdot \text{Cov}(\varepsilon,\varepsilon_f) + \Sigma_x \cdot (\text{Cov}(\varepsilon,\varepsilon_f))^T)\Sigma_x^{-1})}{2(1+r)\text{Tr}(\Sigma_x^{-1}\Sigma_x \cdot \text{Var}(\varepsilon_f)\Sigma_x^{-1})}$$

$$\underset{\text{cov is symmetric}}{=} \frac{\text{Tr}(\Sigma_x^{-1}\Sigma_x \cdot \text{Cov}(\varepsilon,\varepsilon_f)\Sigma_x^{-1})}{(1+r)\text{Tr}(\Sigma_x^{-1}\Sigma_x \cdot \text{Var}(\varepsilon_f)\Sigma_x^{-1})}.$$

This results in the simplified optimal value for $\lambda_{\text{PPI++}}^*$ in the linear case:

$$\lambda_{\text{PPI++}}^* = \frac{1}{1+r} \cdot \frac{\text{Cov}(\varepsilon,\varepsilon_f)}{\text{Var}(\varepsilon_f)}. \tag{18}$$

This demonstrates that for the general linear case with MSE loss, the optimal $\lambda$ depends on the covariance between true $\varepsilon$ and pseudo-labeled $\varepsilon_f$ residuals, the variance of the pseudo-labeled residuals$\text{Var}(\varepsilon_f)$, and the labeled-unlabeled data ratio $r$.

### C.2  PP-SSL expression

From Corollary 3.3, the optimal $\lambda^*$ that minimizes the variance bound is:

$$\lambda^* = \frac{1}{1+r} \cdot \frac{\sigma^2}{\sigma^2 + \sigma_e^2},$$

where $r = n/N$ is the labeled-unlabeled ratio and $\sigma^2$ and $\sigma_e^2$ are constants from Assumption 8, 9.

For linear regression with MSE loss, by assuming independence between features and residuals we have:

$$\sigma^2 = \sup_{w\in\mathbb{R}^d} \mathbb{E}_{x,y\sim P}\|\nabla\ell(w;x,y) - \nabla\mathcal{L}(w)\|^2 = \mathbb{E}_{x,y\sim P}[\varepsilon^2\|x\|^2] = \text{Var}(\varepsilon)\text{Tr}(\Sigma_x),$$

and
$$\sigma_e^2 = \mathbb{E}_{x,y\sim P}(\varepsilon_f^2 \cdot x^\top x - \|\mathrm{Cov}(x,\varepsilon_f)^2\|) = \mathrm{Var}(\varepsilon_f) \cdot \mathrm{Tr}(\Sigma_x).$$

Substituting the two expressions into $\lambda^*_{\text{PP-SSL}}$, yields

$$\lambda^*_{\text{PP-SSL}} = \frac{1}{1+r} \cdot \frac{\mathrm{Var}(\varepsilon)}{\mathrm{Var}(\varepsilon) + \mathrm{Var}(\varepsilon_f)}.$$

The above expression shows that the optimal $\lambda^*_{\text{PP-SSL}}$ depends on the variances of true and pseudo-labeled residuals, and $r$—the ratio between the number of labeled and unlabeled points.

While both `PPI++` and `PP-SSL` down-weight the contribution of the pseudo-labeled data based on the accuracy of the teacher, they differ in key ways. When pseudo-labels are perfect (i.e., $\varepsilon_f = \varepsilon$), both `PPI++` and `PP-SSL` assign high weights to the pseudo-label term: $\lambda^*_{\text{PP-SSL}} = \frac{1}{2(1+r)}$ and $\lambda^*_{\text{PP}} = \frac{1}{1+r}$, with `PPI++` placing more trust in the pseudo-labels. When pseudo-labels are noisy or uncorrelated with the true residuals, both methods reduce the contribution of the pseudo-labeled data by assigning smaller $\lambda^*$ values.

The analysis of `PP-SSL` under the simple linear regression model with MSE loss can be conducted not only through minimizing a bound on the gradient variance, but also by directly minimizing the gradient variance itself. In fact, this direct optimization yields the same analytical solution for $\lambda^*$ as in `PPI++`; see the derivation below. However, the two methods differ in practice: `PPI++` computes gradients and Hessians at the optimal point using all data offline, while `PP-SSL` adapts $\lambda$ dynamically during training by minimizing the variance of the gradient estimate on-the-fly. Consequently, the actual values of $\lambda$ used in training often differ between the two approaches, despite the similar underlying theory. This is reflected in performance differences, as shown in Figure S3.

For completeness, we conclude this discussion by presenting the analytical solution for our $\lambda^*$ that minimizes the prediction-powered gradient variance under a linear model. Recall $g_\lambda^{\text{PPI}}(w)$ from Equation 6, its $\mathrm{Cov}(g_\lambda^{\text{PPI}}(w))$ can be reformulated as

$$\mathrm{Cov}(g_\lambda^{\text{PPI}}(w)) = \frac{\mathrm{Cov}(\nabla\ell)}{n} - \frac{\lambda(\mathrm{Cov}(\nabla\ell,\nabla\ell^f) + \mathrm{Cov}(\nabla\ell^f,\nabla\ell))}{n} + \lambda^2(r+1)\frac{\mathrm{Cov}(\nabla\ell^f)}{n}.$$

For linear regression with MSE loss, assuming independence between features and residuals, as detailed before, the covariance matrices share a common factor $\Sigma_x$, resulting in

$$\mathrm{Cov}(g_\lambda^{\text{PPI}}(w)) = \frac{\Sigma_x}{n}\left(\mathrm{Var}(\varepsilon) - \lambda(2\mathrm{Cov}(\varepsilon,\varepsilon_f)) + \lambda^2(r+1)\mathrm{Var}(\varepsilon_f)\right).$$

By taking the gradient with respect to $\lambda$, equating it to zero, and re-arranging the terms we get:

$$\lambda^*_{\text{linear}} = \frac{1}{1+r}\frac{\mathrm{Cov}(\varepsilon,\varepsilon_f)}{\mathrm{Var}(\varepsilon_f)}.$$

Observe the above $\lambda^*_{\text{linear}}$ coincides with the one from Equation (18).

## D  Synthetic regression experiments with two-groups: supplementary details

In this section, we provide additional experimental details related to the experiments described in Section 4.1. This includes the data generation process and hyperparameter configuration (Section D.1), extended results for the same experimental setup (Section D.2), and a new experiment where the group indicator is assumed to be unknown (Section D.3). Lastly, Section D.4 presents a comparison between models trained with and without access to the group indicator.

### D.1  Experimental details

**Metrics**  We performed tests of solving regression problems, therefore our metrics are as expected for those experiments:

- MSE (Mean Squared Error), defined as $\frac{1}{n_{\text{test}}}\sum_{i=1}^{n_{\text{test}}}(f(x^i) - y^i)^2$, where $f(x)$ is a regression model. (lower is better)
- $R^2$, defined as $1 - \frac{\sum_{i=1}^{n_{\text{test}}}(f(x^i)-y^i)^2}{\sum_{i=1}^{n_{\text{test}}}(\bar{y}-y^i)^2}$, where $\bar{y}$ is the empirical mean of $y$ values in the test set. (higher is better)

**Setup and environment**  The experiments were conducted on a system running Ubuntu 20.04.6 LTS, each experimnt (single seed) with 2 CPU cores of Intel(R) Xeon(R) Gold CPUs at 2.40 GHz, 32 GB of RAM. The software environment used Python 3.11.3 and PyTorch 2.5.1.

**Data generation process**  We design a controlled synthetic regression experiment to evaluate the performance of our method in a setting with group-wise label shift. The dataset consists of $n$ samples $x^i \in \mathbb{R}^m$, where each row is independently drawn from a standard normal distribution $\mathcal{N}(0, I_m)$. The weight vector $w \in \mathbb{R}^m$ is also sampled from $\mathcal{N}(0, I_m)$. The target labels are computed as $y^i = w^\top x^i + b^i$, where $b \in \mathbb{R}^n$ is a bias term drawn from $\mathcal{N}(0, I_n)$. To create two distinct groups, we define a split threshold $\tau = 0.2$, which corresponds to the $\tau$-th percentile of the linear projection $w^\top x$ (excluding the bias term). For samples in the second group (i.e., those above the threshold), we modify the target by adding noise: $y = w^\top x + b + \epsilon$, where $\epsilon \sim \mathcal{N}(\mu, 1)$. By varying $\mu$, we control the shift in label distribution between groups and evaluate the robustness of each method. In the experiments from Section 4.1, which assume access to the group indicator, we augment the generated features with an additional feature representing the group identity. This results in $m + 1$ input features. The ground truth labels are generated using a weight vector $w \in \mathbb{R}^{m+1}$, where the last component—corresponding to the group feature—is zero. This setup allows us to train a model with an expanded input space, enabling it to capture potential group-dependent variations in the data.

**Model details**  We train a linear model in PyTorch with weights of size $m$ (or $m + 1$ when the group indicator feature is included) and a bias term, using the ADAM optimizer.

**Hyperparameters**  Table S1 summarizes the hyperparameters used in the experiments described in Sections D.3, D.2, and 4.1.

Table S1: Experimental settings for synthetic regression and two-groups regression tasks

| Parameter | Synthetic regression without group indicator | Synthetic regression with group indicator |
|---|---|---|
| Total number of samples | 2000 | 2000 |
| Number of features ($m$) | 10 | 10 (+1 group indicator) |
| Split value ($\tau$) | 0.2 | 0.2 |
| Optimizer | Adam | Adam |
| Batch size | 256 | 256 |
| Learning rate | 0.001 | 0.001 |
| Epochs | 3000 | 3000 |
| Number of labeled samples ($n$) | 20 | 20 |
| Number of unlabeled samples ($N$) | 990 | 990 |
| Number of test samples ($n_{\text{test}}$) | 790 | 790 |
| Number of validation samples ($n_{\text{val}}$) | 200 | 200 |
| Labeled fraction | 1% | 1% |
| Unlabeled fraction | 50% | 50% |
| Validation fraction | 10% | 10% |
| Test fraction | 39% | 39% |
| Number of seeds | 100 | 100 |
| Early stopping | Enabled | Enabled |

## D.2   Training with access to the group indicator

This section provides a detailed report of the experiments summarized in Section 4.1. We evaluate performance across a range of noise means $\mu \in [0.1, 7]$, which are applied as a bias to group B. Models are assessed using mean squared error (MSE) and $R^2$ scores, both overall and per group. Figure S1 presents the MSE for all methods across the extended range of noise biases $\mu$, expanding upon the results shown in Figure 1. Figure S2 reports the corresponding $R^2$ scores on the full test set. Note that higher $R^2$ values indicate better performance, while $R^2 < 0$ implies that a naive predictor (i.e., always predicting the mean $\bar{y}$) performs better. Following that figure, we can see that our method matches or outperforms all baselines across the full range of noise values. The performance gap

increases as the bias in group B increases, demonstrating our method's robustness to inaccurate pseudo-labels and its ability to effectively leverage the group indicator.

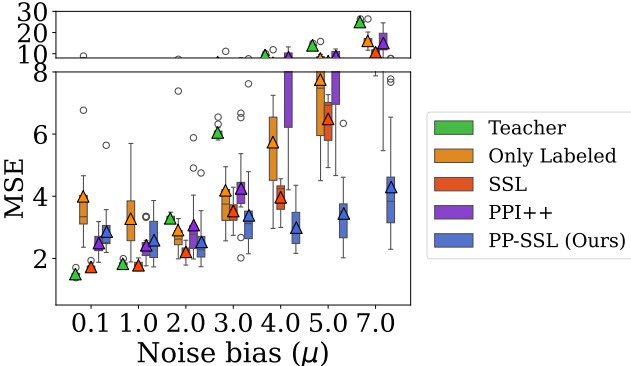

Figure S1: MSE on synthetic data as a function of noise bias $\mu$. Results are shown for the entire test set. Results evaluated on 100 independent experiments.

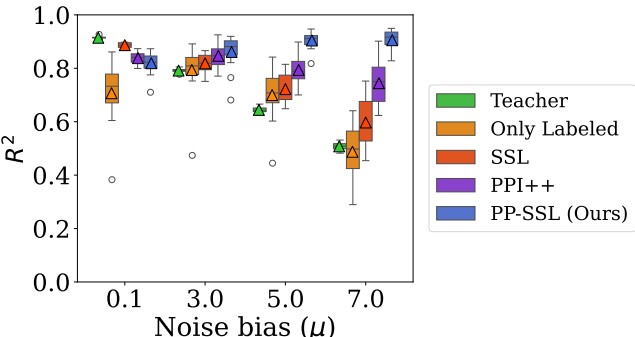

Figure S2: $R^2$ on synthetic data as a function of noise bias $\mu$. Results on the entire test set when training with access to the group indicator feature. Results evaluated on 100 independent experiments.

### D.3 Training without access to the group indicator

This experiment extends the setup from Section 4.1, but differs in a key aspect: it does not assume access to the group indicator for each sample.

The data is generated as described in Section D.1, using the hyperparameters specified in Table S1. Figure S3 shows the group-wise MSE. For group A, where the teacher is accurate, all methods perform similarly well, with the `Teacher` model outperforming others. However, in group B, which is affected by biased noise, we can see the advantage of our approach—especially as the noise mean $\mu$ increases. This highlights our method's robustness even in the absence of explicit group information. As can be seen in Figure S4 our method also achieves the lowest MSE and the highest $R^2$ under high noise-bias conditions across both groups, while maintaining competitive results when noise-bias is low.

Figure S5 presents the MSE curves evaluated on the training and test data. As can be seen, our `PP-SSL` method converges faster than the `Only Labeled` baseline, which is in line with the analysis from Section 3. Observe also that `PPI++` exhibits a similar convergence rate.

### D.4 Effect of group indicator: a comparative analysis

We now provide a side-by-side comparison of the performance of the models trained with and without access to the group indicator. Figure S6 shows that including a group indicator significantly improves

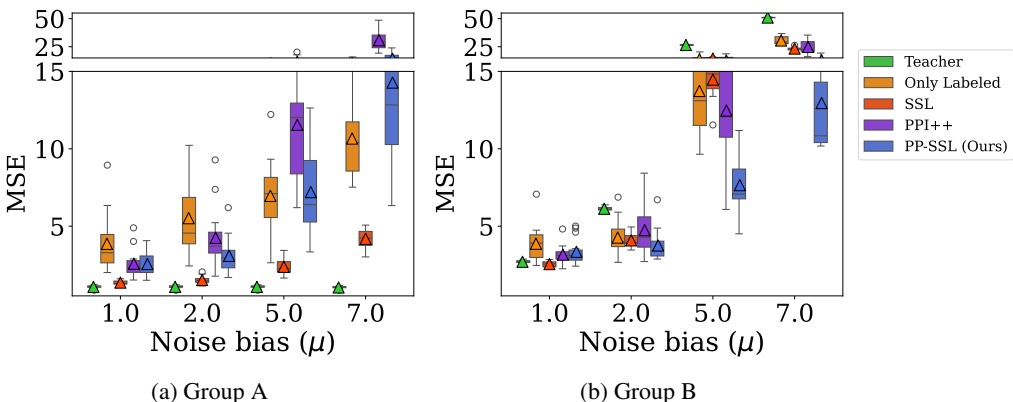

|  | (a) Group A | (b) Group B |
|---|---|---|

Figure S3: MSE on synthetic data without group indicator as a function of noise bias $\mu$. Results are shown for: (a) the full test set; (b) Group A, where the teacher model is accurate (oracle-like); and (c) Group B, which includes additive biased noise. Results evaluated on 100 independent experiments.

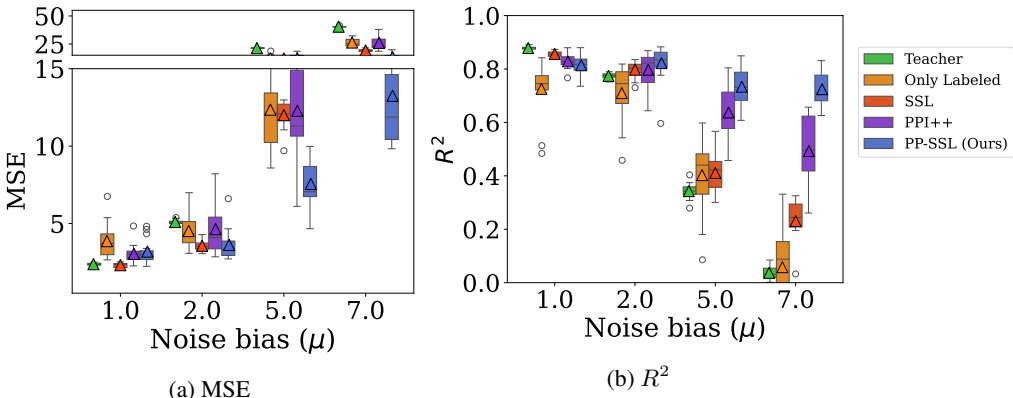

|  | (a) MSE | (b) $R^2$ |
|---|---|---|

Figure S4: Overall MSE and $R^2$ on synthetic data without group indicator, across all test samples versus noise bias $\mu$ added to group B.

model performance, as expected. These results highlight the value of leveraging group information, particularly for methods like ours that can effectively utilize this knowledge.

# E   Tabular real data experiments: supplementary details

In this section, we provide additional experimental details related to the experiments described in Section 4.2. This includes the data split process and hyperparameter configuration (Section E.1), and extended results for the same experimental setup (Section E.2).

## E.1   Experimental details

**Metrics**    We report both MSE and $R^2$ metrics to evaluate model performance.

**Setup and environment**    All experiments were run on an Ubuntu 20.04.6 LTS system. In order to run single seed experiment, hardware included 2 CPU cores from an Intel(R) Xeon(R) Gold processor at 2.40 GHz and 32 GB of RAM. The software environment used Python 3.11.3.

**Data preparation**    We use the California Housing dataset [34], which consists of 20,640 samples and 8 numerical features: median income, house age, average rooms per household, average bedrooms per household, population, average household size, latitude, and longitude. The target variable is median house value. To simulate a natural distribution shift, we partitioned the data into two groups

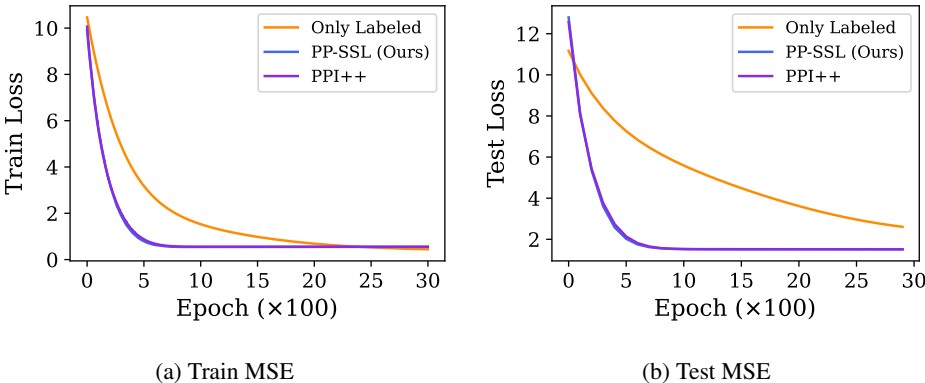

(a) Train MSE          (b) Test MSE

Figure S5: Train and test MSE (loss) during model training using $\mu = 0.1$ for synthetic data experiment.

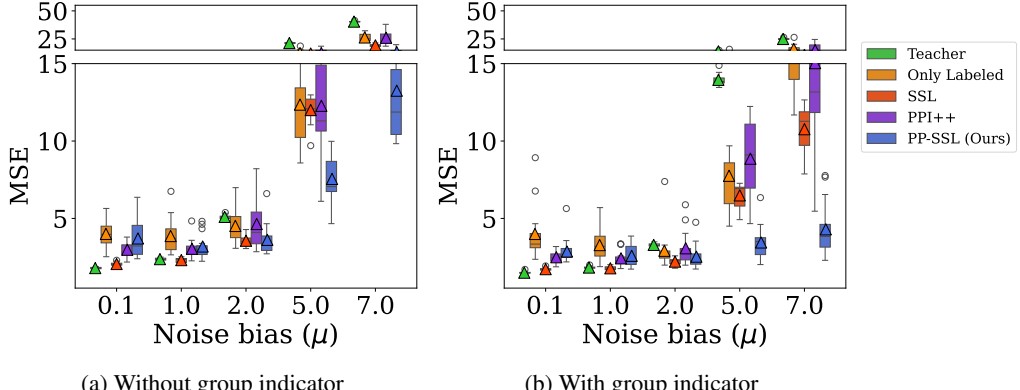

(a) Without group indicator          (b) With group indicator

Figure S6: Synthetic experiments comparison of MSE across different methods as a function of noise level on the entire test set, with and without access to the group indicator feature. **Right:** Models are trained with an additional input feature indicating group membership. **Left:** Models are trained without access to group membership information.

based on the $\tau$-percentile of the target variable $y$. Group A consists of samples with lower target values (bottom $\tau$ percent), and group B contains higher-value samples. We chose $\tau = 0.4$, meaning the bottom $40\%$ of the samples belong to group A. Figure S7 shows the sorted target values used for this grouping.

**Model details** We use a linear model of the form $w^\top x + b$, trained for 1,000 epochs on each configuration. For consistency and fair comparison, all methods (including the `Only Labeled` baseline) were implemented using numpy code, as `PPI++` and our method could not be supported directly by Scikit-learn.

**Hyperparameters** Table S2 summarizes the key hyperparameters and dataset statistics used in the California Housing experiment.

## E.2 Additional results

In what follows, we provide additional results for the experiment from Section 4.2. Figure S9 reports the overall $R^2$ score on the full test set, covering both groups A and B. Figure S8 shows the MSE across the two groups with a broader range of $N_B$ values that were omitted in Section 4.2 for readability. Overall, one can see that the results follow the same trends as those presented in Section 4.2.

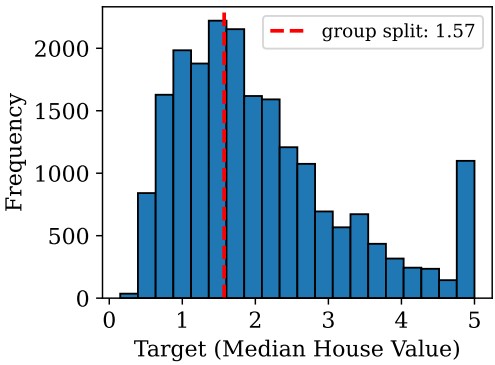

Figure S7: Ordered target house prices illustrating the $40\%$ threshold for group splitting.

Table S2: California Housing Experiment Parameters

| Parameter | Value |
|---|---|
| Number of labeled samples ($n$) | 102 (0.5%) |
| Number of unlabeled samples ($N$) | 16,327 (79%) |
| Pre-train sample count ($N_A + \max N_B$) | 102 (0.5%) |
| Pre-train samples from group A ($N_A$) | 51 |
| Pre-train samples from group B ($N_B$) | {51, 38, 25, 12, 10, 5, 2, 1} |
| Number of test samples ($n_{\text{test}}$) | 4,108 (20%) |
| Number of features ($m$) | 8 |
| Split threshold ($\tau$) | 0.4 |
| Number of seeds | 100 |
| Epochs | 1000 |
| Optimizer | SGD |
| Batch size | Full dataset |
| Learning rate | 0.01 |
| Early stopping | Enabled (patience: 10 steps) |

# F   Visual real data experiments: supplementary details

## F.1   Experimental details

**Metrics**   For evaluation, we report the Mean Average Error (MAE) on the full test set as well as separately on group A (ages 0–30) and group B (ages 31+). MAE is defined as $\frac{1}{n_{\text{test}}} \sum_{i=1}^{n_{\text{test}}} |f(x^i) - y^i|$, where $f(x)$ is a regression model. (lower is better) We report also MSE and $R^2$.

**Setup and environment**   All experiments were conducted on a high-performance computing cluster running Ubuntu 20.04.6 LTS. The hardware configuration includes 98 CPU cores (Intel(R) Xeon(R) Gold 2.40GHz), 256 GB of RAM, and 8 NVIDIA A40 GPUs. The software stack consists of Python 3.11.3, PyTorch 2.5.1, and CUDA 12.4.

**Data preparation**   We use the UTKFace dataset [35], which comprises over 20,000 face images labeled with age, gender, and ethnicity. Images are preprocessed by applying standard face alignment followed by resizing to 128×128 pixels. For data augmentation during training, we apply random horizontal flipping, random rotations, and color jitter to improve generalization. The age labels in UTKFace range from 0 to 116 years, which we treat as a regression problem to predict the exact age.

**Model details**   All models are based on the ResNet-50 architecture pretrained on ImageNet, downloaded from "https://download.pytorch.org/models/resnet50-11ad3fa6.pth". The final classification head is replaced with a fully connected module composed of: Dropout(0.2) $\rightarrow$

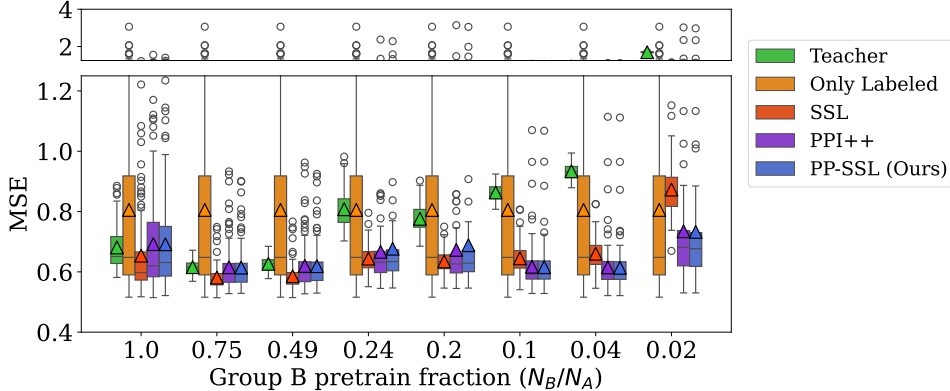

Figure S8: Results on California Housing dataset: MSE as a function of $N_B/N_A$ on the entire test set. Other details are as in Figure 3.

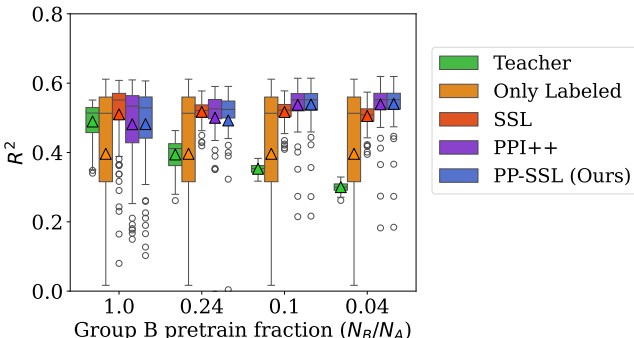

Figure S9: Results on California Housing dataset: $R^2$ as a function of $\frac{N_B}{N_A}$ on the entire test set. Other details are as in Figure 3.

Linear(2048, 256) $\to$ ReLU $\to$ Linear(256, 1), as implemented in the open-source codebase in [40]. This architecture is used across all methods.

**Hyperparameters**   We use the same training setting for all methods, as summarized in Table S3.

### F.2   Additional results

In Section 4.3.1, we present experiments using the MAE performance metric. Here, we complement those by reporting additional results using the MSE and $R^2$ metrics, evaluated both per group and on the entire test set. Figure S10 presents the results based on the MSE metric, while Figure S11 displays the corresponding $R^2$ scores.

Notably, our proposed `PP-SSL` method consistently achieves the best performance among all semi-supervised approaches. It demonstrates noticeable improvements, with particularly strong gains in group B where the `Teacher` model in relatively inaccurate. Additionally, the low variability of the results highlights the stability of `PP-SSL` across different training runs.

## G   CIFAR-10 real data experiments: supplementary details

### G.1   Experimental details

**Metrics**   For evaluation, we report the top-1 Accuracy on the full test set as well as separately on Group A (clean classes) and Group B (corrupted classes). Accuracy is defined as correct-classification by all classifications.

Table S3: Training and experimental configuration for the UTKFace age estimation experiments.

| Parameter | Value |
|---|---|
| Backbone architecture | ResNet-50 pretrained on ImageNet |
| Model final layers | Dropout(0.2) $\rightarrow$ Linear(2048,256) $\rightarrow$ ReLU $\rightarrow$ Linear(256,1) |
| Loss function | L1 |
| Optimizer | SGD |
| Momentum | 0.9 |
| Weight decay | 0.001 |
| Initial learning rate | 0.001 |
| Batch size | 512 |
| Epochs | 100 |
| Early stopping | Patience = 10 (based on validation loss) |
| Number of seeds | 5 |
| Pretrain set fraction | 7% (1,628 samples) |
| Labeled set fraction | 3% (698 samples) |
| Unlabeled set fraction | 70% (16,288 samples) |
| Validation set fraction | 10% (2,326 samples) |
| Test set fraction | 10% (2,062 samples) |
| Group A size (ages 0–29) | 813 samples |
| Group B sizes (ages 30+) | $\{16, 33, 81, 163, 325, 488, 816\}$ |

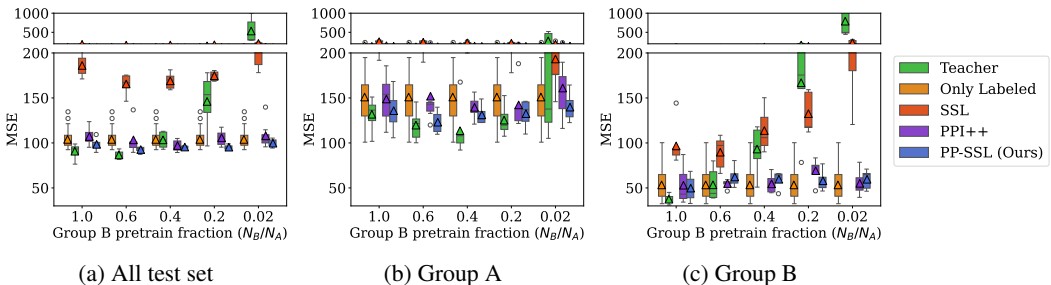

(a) All test set      (b) Group A      (c) Group B

Figure S10: Results for age estimation: MAE as a function of $N_B/N_A$, across 5 data splits. Other details are as in Figure 4

**Setup and environment** All experiments were conducted on a high-performance computing cluster running Ubuntu 20.04.6 LTS. The hardware configuration includes 98 CPU cores (Intel(R) Xeon(R) Gold 2.40GHz), 256 GB of RAM, and 8 NVIDIA A40 GPUs. The software stack consists of Python 3.11.3, PyTorch 2.5.1, and CUDA 12.4.

**Data preparation** We use the CIFAR-10 dataset [37], a standard benchmark comprising 60,000 color images in 10 classes (airplane, automobile, bird, cat, deer, dog, frog, horse, ship, and truck), with 6,000 images per class. The dataset is split into 50,000 training images and 10,000 test images. For our experiment, we reserve 5,000 images from the training set to pretrain a teacher model, and the remaining 45,000 images are used for student training.

All images are resized to $256 \times 256$ and center-cropped to $224 \times 224$ to match the input requirements of pretrained ResNet-50 models. We apply standard data augmentation for training, including random resized crops and horizontal flipping. To evaluate model robustness, we optionally add visual corruptions (e.g., Gaussian noise, saturation, brightness) to selected classes such as *cat*, *dog*, and *deer* using a class-aware corruption transform prior to normalization.

**Model details** We extract features using a ResNet-50 model pretrained on ImageNet, obtained from the official PyTorch model zoo at `https://download.pytorch.org/models/resnet50-11ad3fa6.pth`. The ResNet encoder outputs a 2048-dimensional feature vector for each image. We replace the final classification head with a single linear layer of shape (2048, 10) to classify among the 10 CIFAR-10 classes.

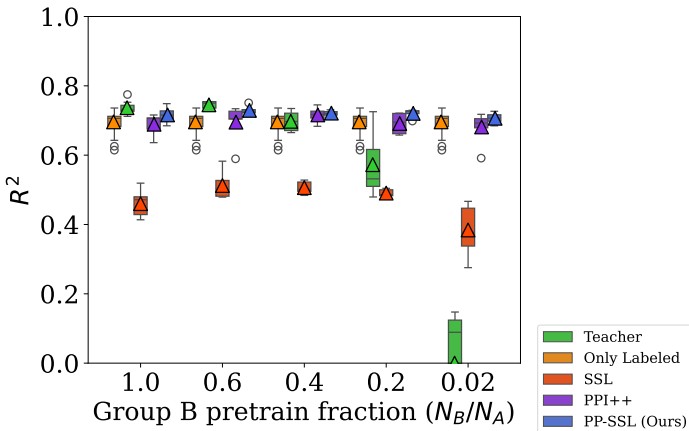

Figure S11: Results for age estimation: $R^2$ as a function of $N_B/N_A$, on the entire test set across 5 data splits. Other details are as in Figure 4

In knowledge distillation settings, we initialize the teacher with a pretrained linear model and load the checkpoint into a duplicate of the student architecture. The teacher is frozen during training and provides soft targets for distillation-based supervision.

**Hyperparameters**   Table S4 summarizes the key hyperparameters and dataset statistics used in the California Housing experiment.

Table S4: California Housing Experiment Parameters

| Parameter | Value |
| --- | --- |
| Number of labeled samples ($n$) | 4,500 (9%) |
| Number of unlabeled samples ($N$) | 40,500 (81%) |
| Number of pretrain samples ($N_{pre}$) | 5,000 (10%) |
| Number of test samples ($n_{\text{test}}$) | 10,000 |
| Number of features in classifier | 2048 |
| Number of seeds | 5 |
| Epochs | 100 |
| Optimizer | SGD |
| Batch size | 256 |
| Learning rate | 0.001 |
| Labeled Loss | CE |
| Unlabeled Loss | KL divergence |

### G.2   Experimental Setup

In this section, we present an empirical study on semi-supervised learning under class-conditional corruption, using the CIFAR-10 dataset as a controlled testbed. Our goal is to investigate how the presence of visual corruption in a subset of classes affects learning, and how prediction-powered or distillation-based methods can mitigate these effects. To that end, we inject various types of corruption into the dataset in the image space (e.g., saturation, brightness, Gaussian noise). We split the training data into "noisy" and "clean" subsets based on class labels, and study how different training strategies perform in this mixed-quality regime.

Following recent semi-supervised learning pipelines, we first extract 2048-dimensional features from all training and test images using a fixed ResNet50 backbone pretrained on ImageNet. To enable a teacher-student framework, we first split the training set into two parts. A disjoint subset of 5,000 clean training samples is used to train a teacher model. This model is trained in a supervised manner using the standard cross-entropy loss. Once trained, the teacher model is saved and used to guide

downstream learning via knowledge distillation. We evaluate performance separately on corrupted and non-corrupted classes to understand the strengths and limitations of each method under real-world noise scenarios.

To simulate a semi-supervised learning scenario, we split the feature-extracted training data such that only 10% of samples have labels, and the remaining 90% are treated as unlabeled. A linear classifier is trained on this data, with supervision provided via hard labels and, when available, soft predictions from the teacher model. The loss function combines cross-entropy for labeled data with a Kullback–Leibler divergence term for distillation (pseudo-labeled data):

$$\mathcal{L}_{\text{SSL}} = \mathcal{L}_{\text{CE}} + \lambda \cdot \mathcal{L}_{\text{KL}}, \qquad \mathcal{L}_{\text{PP-SSL}} = \mathcal{L}_{\text{CE}} + \lambda \cdot (\mathcal{L}^f_{N,\text{KL}} - \mathcal{L}^f_{n,\text{KL}}),$$

where for a sample with true label vector $y \in \{0,1\}^C$ and predicted probabilities $p \in [0,1]^C$, the cross-entropy loss is defined as:

$$\mathcal{L}_{\text{CE}}(p, y) = -\sum_{i=1}^{C} y_i \log(p_i),$$

where $C$ is the number of classes and $p_i$ is the predicted probability for class $i$. When applying knowledge distillation, we further incorporate a KL divergence loss between the probability distribution produced by the teacher model and that of the student model. The KL divergence is given by:

$$\mathcal{L}_{\text{KL}}(q\|p) = \sum_{i=1}^{C} q_i \log\left(\frac{q_i}{p_i}\right),$$

where $q$ is the distribution from the teacher, and $p$ is the student's predicted distribution.

## G.3 Additional Results

Table S5: Results for corruption severity = 1

|  | Labeled | SSL | PPI++ | PP–SSL | Teacher |
|---|---|---|---|---|---|
| **Brightness** | | | | | |
| Total | 81.42±0.03 | 81.48±0.06 | 77.92±0.02 | 81.23±0.11 | **85.68±0.01** |
| Group A | 85.83±0.03 | 85.97±0.05 | 81.99±0.04 | 84.93±0.06 | **88.50±0.03** |
| Group B | 3.72±0.74 | 4.92±1.41 | 56.93±0.12 | **60.55±0.26** | 4.81±0.30 |
| **Saturate** | | | | | |
| Total | 74.89±0.09 | 74.65±0.30 | 70.82±0.14 | 74.97±0.65 | **79.15±0.06** |
| Group A | 79.11±0.35 | 78.88±0.97 | 81.14±0.03 | **84.12±0.21** | 75.46±0.09 |
| Group B | 88.62±0.02 | **88.70±0.03** | 85.49±0.03 | 88.24±0.04 | 88.53±0.03 |
| **Shot Noise** | | | | | |
| Total | 84.22±0.07 | 84.41±0.19 | 80.96±0.03 | 83.91±0.18 | **88.48±0.03** |
| Group A | 57.44±1.12 | 56.38±3.20 | 70.99±0.10 | **74.41±0.55** | 45.31±0.26 |
| Group B | 78.21±0.33 | 78.45±0.45 | 80.58±0.03 | **83.56±0.09** | 74.45±0.07 |
| **Spatter** | | | | | |
| Total | 82.31±0.03 | 82.34±0.05 | 80.16±0.03 | **83.10±0.06** | 82.86±0.02 |
| Group A | 88.40±0.04 | **88.52±0.06** | 85.48±0.03 | 88.28±0.10 | 88.50±0.03 |
| Group B | 54.36±1.10 | 54.95±1.52 | 69.18±0.09 | **72.57±0.21** | 42.06±0.23 |
| **Speckle Noise** | | | | | |
| Total | 74.08±0.11 | 73.89±0.17 | 75.88±0.06 | **78.82±0.09** | 69.81±0.10 |
| Group A | 63.15±0.22 | 63.57±0.42 | 76.92±0.03 | **79.93±0.09** | 63.19±0.07 |
| Group B | 88.43±0.03 | **88.53±0.02** | 85.46±0.03 | 88.27±0.04 | **88.53±0.03** |

Table S6: Results for corruption severity = 5

|  | Labeled | SSL | PPI++ | PP-SSL | Teacher |
|---|---|---|---|---|---|
| **Brightness** | | | | | |
| Total | 82.89±0.07 | 82.87±0.19 | 80.97±0.17 | **84.26±0.11** | 81.14±0.06 |
| Group A | 72.10±0.28 | 71.63±0.75 | 72.65±0.63 | **77.59±0.38** | 64.15±0.17 |
| Group B | 87.52±0.03 | 87.68±0.09 | 84.54±0.14 | 87.12±0.04 | **88.48±0.06** |
| **Saturate** | | | | | |
| Total | 80.73±0.09 | 80.67±0.13 | 79.95±0.06 | **83.32±0.02** | 77.15±0.15 |
| Group A | 62.94±0.28 | 62.61±0.41 | 67.92±0.19 | **72.94±0.10** | 51.07±0.52 |
| Group B | 88.35±0.02 | 88.41±0.02 | 85.11±0.07 | 87.77±0.03 | **88.48±0.04** |
| **Shot Noise** | | | | | |
| Total | 79.29±0.20 | 79.31±0.45 | 78.92±0.09 | **81.99±0.16** | 77.34±0.14 |
| Group A | 57.93±0.65 | 57.67±1.44 | 63.57±0.19 | **67.31±0.39** | 51.87±0.44 |
| Group B | 88.45±0.03 | **88.58±0.06** | 85.49±0.09 | 88.28±0.10 | 88.50±0.05 |
| **Spatter** | | | | | |
| Total | 66.09±0.42 | 66.13±0.58 | 76.75±0.05 | **79.20±0.11** | 65.46±0.26 |
| Group A | 13.92±1.39 | 13.77±1.95 | 56.37±0.23 | **58.09±0.34** | 12.63±0.91 |
| Group B | 88.45±0.03 | 88.58±0.03 | 85.49±0.05 | 88.25±0.04 | **88.67±0.06** |
| **Speckle Noise** | | | | | |
| Total | 77.69±0.23 | 77.86±0.31 | 78.52±0.08 | **81.53±0.11** | 76.30±0.15 |
| Group A | 52.62±0.74 | 52.95±1.02 | 62.33±0.21 | **65.79±0.31** | 48.13±0.45 |
| Group B | 88.44±0.03 | 88.54±0.02 | 85.46±0.06 | 88.27±0.04 | **88.65±0.05** |

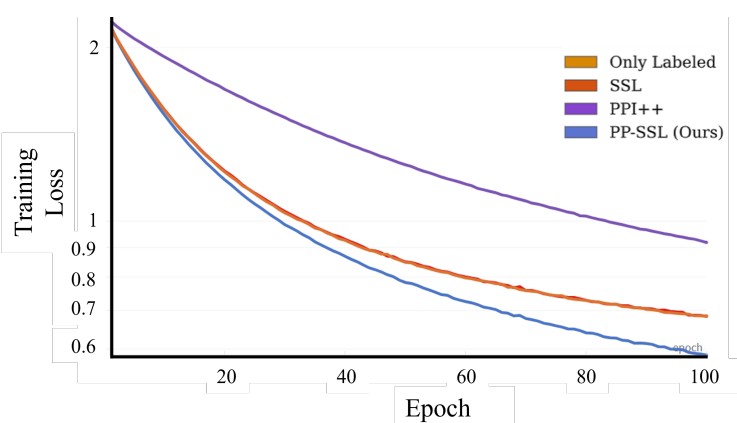

Figure S12: Convergence graph training loss as a function of epochs during CIFAR-10 training

**Analysis.** Tables S5, S6 shows that our method improves the accuracy on the corrupted classes across different corruption severities, with the adaptive choice of $\lambda$ leading to improved performance compared to baselines. The results confirm the robustness of PP-SSL to varying levels of noise.

In addition to accuracy, in Figure S12 we also evaluated convergence time. PP-SSL converges faster than the only-labeled baseline, standard SSL, and PPI++, which require more training epochs to stabilize. This faster convergence further highlights the efficiency of our adaptive weighting strategy in practice.

# H Additional Experiments on the Choice of $\lambda$

In this section, we provide additional experiments that validate our theory regarding the optimal weight parameter $\lambda$. These results complement the insights derived in Corollary 3.3.

## H.1 Synthetic Regression: Theoretical vs. Empirical $\lambda^*$

We first conduct a controlled synthetic regression experiment to compare the theoretical $\lambda^*$ from Corollary 3.3 (i.e., in Eq. 10) with the empirical value that minimizes the gradient variance. Since the theoretical $\lambda^*$ (denotes as $\lambda^*_{\text{theory}}$ below) is derived by minimizing an *upper bound* on the variance rather than the variance itself, it need not coincide exactly with the empirical variance minimizer (denoted as $\lambda^*_{\text{practice}}$).

**Experimental setup.** Synthetic data is generated according to the linear noisy model $y = \mathbf{x}^\top \mathbf{w}^* + \epsilon$, with $\mathbf{x} \sim \mathcal{N}(0, I_d)$ where $d = 10$, $\epsilon \sim \mathcal{N}(0, \sigma_*^2)$ where $\sigma_*^2 = 0.01$, and $\mathbf{w}^* \sim \mathcal{N}(0, I_d)$. We sample $n = 50$ labeled examples and $N \in \{200, 2000\}$ unlabeled examples from this model. A synthetic teacher provides pseudo-labels of the form $f(\mathbf{x}) = \mathbf{x}^\top(\mathbf{w}^* + b\mathbf{e}_1) + \zeta$, where $b \in \mathbb{R}$ is a bias parameter and $\zeta \sim \mathcal{N}(0, \sigma_\zeta^2)$ introduces noise. Both $b$ and $\sigma_\zeta^2$ control the teacher's quality. Since directly mapping them to $\sigma^2$ and $\sigma_e^2$ is non-trivial, we set $b \in \{0.1, 1\}$, $\sigma_\zeta^2 \in \{0.01, 1.5\}$ and empirically estimate $\sigma^2$ and $\sigma_e^2$. We consider the MSE loss $\ell(w; x, y) = \frac{1}{2}(x^\top w - y)^2$ and estimate its gradient at a randomly sampled test point $\tilde{w} \sim \mathcal{N}(0, I_d)$.

**Results.** Table S7 reveals that $\lambda^*_{\text{theory}}$ consistently approaches the performance (in terms of measured variance) of $\lambda^*_{\text{practice}}$ across different settings. The theoretical value yields variance reductions within a narrow margin of the empirical optimum.

Table S7: Comparison between theoretical and empirical $\lambda^*$. Theoretical values match empirical ones closely in terms of minimizing gradient variance.

| $r$ | $\sigma_\zeta^2$ | $b$ | $\sigma^2$ | $\sigma_e^2$ | $\lambda^*_{\text{theory}}$ | $\lambda^*_{\text{practice}}$ | $\text{Var}(\lambda^*_{\text{theory}})$ | $\text{Var}(\lambda^*_{\text{practice}})$ |
|---|---|---|---|---|---|---|---|---|
| 0.025 | 0.01 | 0.1 | 10.4078 | 0.0793 | 0.9682 | 0.99 | 0.0050 | 0.0046 |
| 0.025 | 1.5 | 1 | 14.1715 | 7.6479 | 0.6336 | 0.70 | 0.1733 | 0.1669 |
| 0.25 | 0.01 | 0.1 | 11.5082 | 0.0440 | 0.7969 | 0.77 | 0.0749 | 0.0709 |
| 0.25 | 1.5 | 1 | 11.9349 | 10.1486 | 0.4324 | 0.35 | 0.1034 | 0.1004 |

## H.2 Adaptive Dynamics of $\lambda$

Next, we investigate how the adaptive algorithm adjusts $\lambda$ over training epochs in relation to teacher quality. We track $\lambda_t$ when initialized at $\lambda_0 = 1$ across 1000 epochs for teachers of varying pseudo-label mean squared error (MSE).

**Results.** Table S8 shows that $\lambda_t$ converges automatically toward the theoretical $\lambda^*$. Moreover, the limiting value of $\lambda$ increases as the teacher error decreases, which matches intuition: better teachers justify a higher weight on pseudo-labels.

Table S8: Adaptive dynamics of $\lambda$ across epochs. $\lambda_t$ converges toward the theoretical $\lambda^*$, with larger $\lambda$ for higher-quality teachers.

| Teacher MSE | $\lambda_0$ | $\lambda_{100}$ | $\lambda_{200}$ | $\lambda_{300}$ | $\lambda_{400}$ | $\lambda_{500}$ | $\lambda_{600}$ | $\lambda_{700}$ | $\lambda_{800}$ | $\lambda_{900}$ | $\lambda_{1000}$ | $\lambda^*_{\text{theory}}$ |
|---|---|---|---|---|---|---|---|---|---|---|---|---|
| 0.01 | 1.000 | 0.966 | 0.946 | 0.934 | 0.929 | 0.926 | 0.925 | 0.924 | 0.924 | 0.924 | 0.924 | 0.968 |
| 0.04 | 1.000 | 0.945 | 0.902 | 0.866 | 0.836 | 0.811 | 0.791 | 0.775 | 0.761 | 0.751 | 0.742 | 0.795 |
| 0.05 | 1.000 | 0.930 | 0.867 | 0.812 | 0.766 | 0.727 | 0.696 | 0.671 | 0.651 | 0.635 | 0.622 | 0.634 |

### H.3 Comparison to the Infeasible Best Fixed $\lambda$

Finally, we benchmark our adaptive method (PP-SSL) against an oracle baseline that uses the infeasible "best" fixed $\lambda$ minimizing test error. We employ the subgroup bias synthetic regression setup from Section 4.1.

**Results.** Table S9 shows that PP-SSL matches the performance of the infeasible oracle across all bias levels. This demonstrates that adaptivity is sufficient to recover the performance of the best fixed $\lambda$, without prior knowledge of the optimal value.

Table S9: Comparison of PP-SSL to the (infeasible) best $\lambda$. Our adaptive algorithm matches the oracle performance across all bias settings.

| Bias $\mu$ | Inf. $\lambda^*$ | Inf. MSE (Total/A/B) | PP-SSL $\lambda$ | PP-SSL MSE (Total/A/B) |
|---|---|---|---|---|
| 0.1 | 0.50 | 1.93 / 1.76 / 2.59 | 0.46 | 1.93 / 1.77 / 2.58 |
| 0.5 | 0.50 | 1.85 / 1.69 / 2.50 | 0.59 | 1.85 / 1.69 / 2.50 |
| 3.0 | 0.35 | 2.76 / 1.95 / 6.05 | 0.33 | 2.76 / 1.96 / 6.04 |
| 5.0 | 0.35 | 2.45 / 2.07 / 4.05 | 0.36 | 2.47 / 2.06 / 4.12 |

**Summary.** Across all three settings, we observe strong alignment between the theoretical prescription, the adaptive algorithm, and the empirical optimum. This provides further evidence that the proposed choice of $\lambda$ is both theoretically sound and practically effective.

