# OpenReview forum: "Prediction-Powered Semi-Supervised Learning with Online Power Tuning"
_NeurIPS.cc/2025/Conference — NeurIPS 2025 poster_

### Official Review · Reviewer_Gh11 · 2025-06-29

**Clarity:** 3
**Significance:** 3
**Originality:** 3
**Rating:** 6
**Confidence:** 4

**Summary:**

This paper proposes Prediction-Powered Semi-Supervised Learning (PP-SSL), which is a fresh direction for dealing with the classic challenge in semi-supervised learning: how to get the most out of many unlabeled samples when the labeled ones are scarce and expensive. Traditional approaches, for example simple pseudo-labeling or teacher-student frameworks, often run into trouble when the teacher model makes mistakes—especially for minority groups. Such errors in pseudo-labels can bias the learning process, causing the trained model to perform worse than one would expect, or even worse than using only the labeled data.

The authors build on the Prediction-Powered Inference (PPI) principle, where pseudo-labels are used both for labeled and unlabeled data in a way that the expected bias cancels out. In earlier work, the weighting between real and pseudo-labels—this λ parameter—was set offline, based on quantities like the variance of the labeled data and the accuracy of the teacher model. But these are usually unknown in practice.

What makes PP-SSL special is the automatic, online adjustment of λ during training, using the AdaGrad algorithm. With this, the method finds a good balance between using real labels and pseudo-labels on the fly, without needing prior knowledge or manual tuning. The paper provides a theoretical analysis, showing that the new gradient estimator is unbiased and can achieve lower variance than the standard supervised gradients, leading to faster and more stable training, especially as the accuracy of the teacher improves or more unlabeled data becomes available.

The authors give thorough experiments, both on synthetic datasets (with carefully designed group imbalances) and on real-world datasets like California Housing and UTKFace for age estimation. The results show that PP-SSL consistently matches or outperforms standard SSL and previous PPI-based methods, especially in difficult cases where the teacher model is weak for certain subgroups. The approach also proves more robust in situations with distribution shifts between groups.

Finally, the paper openly discusses limitations, such as the assumption that labeled and unlabeled samples are from the same distribution, and the current focus on fixed (not self-updating) teacher models. There is also attention to broader societal impacts, including fairness and robustness, and the need for further work to handle distributional shifts or to apply the method to more task types.

In summary, PP-SSL gives a principled, practical improvement for semi-supervised learning, making better use of unlabeled data while protecting against the classic pitfall of pseudo-label bias, especially in real-world, unbalanced datasets.

**Questions:**

1. The authors briefly mention self-training as an alternative to the teacher-student approach in SSL. Given that they propose an *online* learning approach within the teacher-student paradigm, onw wonders whether they can transfer their ideas to self-training (which is online by conceptualizatoin)? In particular, could you use the same model for both labeled and pseudo-labeled data, but iteratively update pseudo-labeled data only, while dynamically tuning λ as in PP-SSL to continuously adapt how much weight to give to pseudo-labels. In this case, could some of the convergence results be translated to self-training?

2. "A key limitation of our method, shared by many SSL methods, is the assumption that labeled and unlabeled data are drawn from the same distribution. ..... Extending our framework to handle such shifts is a natural direction for future work." Can the authros share some ideas of how thery are aiming to do that in the PP-SSL framework? https://www.jmlr.org/papers/v23/22-0541.html or (more general) https://arxiv.org/pdf/2505.07367 might be starting points for the iterative/online case.

**Ethical Concerns:**

["NO or VERY MINOR ethics concerns only"]

**Final Justification:**

see reply to rebuttal

**Limitations:**

yes

**Paper Formatting Concerns:**

Reference [2] misses a publisher or journal or url.

**Quality:**

4

**Strengths And Weaknesses:**

The method reduces bias from bad pseudo-labels by tuning the λ parameter automatically during training, so you do not have to guess the right value beforehand. It works well even if the teacher model is not perfect, especially for smaller groups that often get ignored. The approach is backed by clear theoretical results, and also shows strong improvements in practical experiments with real and synthetic data. The algorithm is simple to use with existing optimizers like AdaGrad, so implementation is straightforward.

The main weakness, as the auhtors themselves note, is that the method assumes the labeled and unlabeled data are coming from the same distribution. As a minor weakness imo, most experiments are in regression settings. Considering classification tasks too would make the results stronger.

---

> ### Author Rebuttal · Authors · 2025-07-30
>
> Thank you for your thoughtful and constructive review. We appreciate your detailed summary and recognition of both the theoretical contributions and practical strengths of our work. Your comments about the novelty of dynamically tuning λ, the robustness to teacher bias, and the simplicity of integrating PP-SSL into existing optimization pipelines are especially encouraging.
>
> Several of your questions overlap with those raised by other reviewers (e.g., Reviewer sxXE), and for convenience, we restate and adapt the relevant parts of our responses here, while providing additional clarifications specific to your suggestions.
>
> ## C1: Could the method be extended to self-training?
> This insightful question was also raised by Reviewer sxXE (C1), and we summarize our response here for your convenience.
>
> In principle, our method can indeed be applied in self-training settings, where the same model is used to generate pseudo-labels. However, this introduces theoretical challenges: the pseudo-label generator is no longer fixed and becomes dependent on the training data and dynamics. As a result, the independence assumptions in our theoretical analysis no longer hold.
>
> Nevertheless, if the teacher is updated using data disjoint from the training set (e.g., a held-out validation set), then the PP-SSL framework and its guarantees can still be extended. In this case, $\sigma_e^2$ can be replaced with a worst-case bound across teacher updates. A more refined treatment would require analyzing a time-varying $\lambda_t^*$ corresponding to each teacher, shifting from static to dynamic regret analysis. This is an exciting direction for future research, and our work provides a principled foundation towards this goal.
>
> ## C2 : How can the method handle distribution shift between labeled and unlabeled data?
> This important point aligns with our discussion of limitations in the paper and was also mentioned by Reviewer sxXE (C5).
>
> While PP-SSL currently assumes a shared distribution between labeled and unlabeled data, we believe the framework can be extended. For example, as done in PPI and Doubly-robust self-training (Zhu et al., 2023), importance weighting can be introduced to correct for covariate shift. This involves estimating the density ratio between labeled and unlabeled covariates and adjusting the loss accordingly. In our context, this could also be done by reweighting the gradient estimator used for λ adaptation, or by modifying the loss function itself to account for distribution mismatch.
>
> Extending our work to handle distribution shifts unlocks the opportunity to use generative models to produce large amounts of unlabeled data from a different distribution. While this may increase the quantity and diversity of available data, it introduces shifts that go beyond simple covariate differences. In such cases, standard importance weighting may not be sufficient to correct for the resulting shift. Addressing this setting would likely require adapting both the pseudo-labeling mechanism and the gradient-based weighting strategy in PP-SSL to account for more complex forms of distribution shift.
>
> We appreciate the suggestion to consider iterative or online extensions of our PP-SSL framework. As the reviewer notes, iterative SSL methods such as the one in [1] offer a natural way to incorporate distributional shifts by sequentially adapting the model on new chunks of unlabeled data. We believe this direction can be complementary to our current work. One potential extension is designing a double-loop algorithm inspired by [1]: the unlabeled data is split into $K$ chunks; for each outer iteration $k = 0,\ldots,K$, we would run our PP-SSL algorithm using the labeled data and the $k$-th chunk of unlabeled data (with a suitable teacher); the resulting model is then used as a teacher to label the next chunk. This setup introduces a time-varying error variance $\sigma\_{e,t}$, which presents new challenges for analysis, particularly in tracking how the optimization dynamics evolve over outer iterations. Extending our convergence results to this setting is a promising direction for future work.
>
> We also reviewed the more recent reciprocal learning framework [2], which introduces an alternating scheme between sample adaptation and ERM updates. However, both [1] and [2] abstract away the optimization process, assuming access to an exact ERM solution or working under strong convexity and Lipschitz assumptions. In contrast, our work explicitly models the training dynamics and optimization errors in non-convex settings, which we believe is crucial for practical SSL scenarios involving deep models.
>
> We agree that incorporating ideas from these works into our framework—while preserving our focus on optimization dynamics—would be a meaningful direction. In future work, we plan to explore how concepts like sample adaptation and staged learning can be integrated into PP-SSL in a way that remains faithful to the stochastic, non-convex nature of modern training procedures.
>
> We will elaborate on these ideas in the final version of the paper’s discussion section.
>
> ## C3: Considering classification tasks would strengthen the results.
>  This suggestion is shared with Reviewers sxXE (C4) and yHkN, and we thank you all for encouraging this extension.
> As part of our ongoing experiments, we applied PP-SSL to a semi-supervised classification task using CIFAR-10 with class-conditional corruption (based on CIFAR-10-C). Our method consistently outperformed strong baselines (SSL, PPI++, DR-ST) in total accuracy and in the corrupted subset (Group A), while maintaining comparable performance on clean classes (Group B). The performance gap on Group A depends on the corruption type and aligns with our theory regarding the pseudo-label error $\sigma_e^2$. We conducted additional tests across multiple corruption types and severity levels and observed the same trends, though we present only severity 5 in the main paper for clarity.
>
> ## C4: Reference formatting
>
> Thank you for catching this. We will correct the missing publisher/journal information for Reference [2] in the camera-ready version.
>
> Once again, thank you for your positive evaluation and for highlighting both the novelty and potential impact of our method. Your feedback has helped us clarify our contributions and better communicate the broader applicability of PP-SSL.
>
> ## References:
> [1] Information-Theoretic Characterization of the Generalization Error for Iterative Semi-Supervised Learning. He et al., JMLR, 2022.
>
> [2] Generalization Bounds and Stopping Rules for Learning with Self-Selected Data.  Rodemann et al.,  2025.

---

> > ### Comment · Reviewer_Gh11 · 2025-08-01
> >
> > Thank you for your detailed answers to all 4 reviewrs. Your clarification regarding the novel aspects, particularly the finite-time analysis and adaptive tuning of λ, is insightful ( response to Reviewer Gh11). Your justification for the theoretical use of the Lipschitz assumption is - in my opinion - appropriately contextualized. I also appreciate the additional empirical validations, especially the extension to widely-used benchmaks like CIFAR-10 and CIFAR-10-C. I think they further strengthen the practical significance of your contributions (as discussed in responses to Reviewers sxXE and yHkN).. Similarly, your response on handling distribution shift adds depth: the ideas of importance weighting, reweighting gradient estimators, and the proposed double-loop iterative scheme are concrete and promisng directions. It would further improve clarity if the manuscript briefly summarized these planned extensions in the limitations or discussion section, perhaps noting what technical obstacles remain for non-stationary teachers or more complex shifts. Two small suggestions: 1) since the CIFAR-10-C experiments play a key role in addressing concerns about applicability to classification, including a concise visualization or summary of how λ behaves under different corruptions would help tie the theory to empirical behavior. 2) Not all the bounds in the recirprocal setup [2] require convexity, see e.g. Thm. 17 in [2]. So remaining "faithful to the stochastic, non-convex nature of modern training procedures" is feasible if you would like to extend to label-unlabel shifts. Overall, your clarifications, added experiments, and forward-looking extensions substantially bolster the paper, though a bit more articulation of the remaining challenges would make the final version even stronger. Again, I appreciate the very detailed and thoughtfoul rebuttal! Thanks.

---

> > > ### Author Response · Authors · 2025-08-04
> > >
> > > We sincerely thank the reviewer for the thoughtful and encouraging feedback.
> > >
> > > We appreciate your recognition of our theoretical and empirical contributions, as well as your helpful suggestions. In the final version, we will incorporate (1) a concise discussion of remaining challenges in handling distribution shifts and non-stationary teachers, and (2) a visualization showing how λ evolves under different CIFAR-10-C corruptions.
> > >
> > > Thank you also for pointing out the broader applicability of the bounds in [2]; we will clarify this in our revised manuscript. Your comments have helped us further improve the clarity and completeness of our work.

---

### Official Review · Reviewer_yHkN · 2025-07-02

**Clarity:** 3
**Significance:** 2
**Originality:** 3
**Rating:** 3
**Confidence:** 4

**Summary:**

The paper proposes using prediction-powered inference (PPI) with online power tuning as a method for training models in a semi-supervised fashion. The basic idea is to optimize the PPI++ loss while running an online learning algorithm to set the power tuning parameter. This method is shown to outperform natural baselines on several datasets.

**Questions:**

- In line 4: What do you mean by a factor of 4 being asymptotically equivalent? Having the same dependence on n? Also, where does the factor 4 come from?

- In line 180: "In contrast, PP-SSL provides a principled mechanism for leveraging unlabeled data that improves statistical efficiency without compromising correctness." What is correctness here? Don't we just care about optimizing the loss? If so, why do we care about not introducing bias?

- Why did you decide to use AdaGrad (as opposed to a different stochastic optimization method)? Did you try any others?

- I don't fully understand what the baseline methods are. What is "teacher model f used for pseudo-labeling"? Is this using f to label all unlabeled data and training a model on everything? Also, what is $\lambda_{\mathrm{PP}}^*$, seeing that your gradients depend on the current iterate (suggesting the tuning parameter should change at every step), unlike in PPI++ where there is a single distribution?

**Ethical Concerns:**

["NO or VERY MINOR ethics concerns only"]

**Final Justification:**

This paper feels borderline. I would feel okay seeing it accepted, but I still think it is lacking empirical evaluation. Its core claim is that it offers a strong baseline for semi-supervised learning. This is the positioning in the abstract and throughout. But we are not seeing non-trivial baselines for semi-supervised learning in the comparisons.

It seems that the authors feel that the theory is an important part of the contribution. I'm sympathetic to that, but then the core messaging should be revised. Overall I think online power tuning is a good idea.

**Limitations:**

Yes

**Quality:**

2

**Strengths And Weaknesses:**

Online power tuning is a very reasonable idea and finding ways to leverage it in a learning context (in combination with PPI debiasing techniques) is a great direction.

My main concern is about whether the empirical evaluation is sufficient. Semi-supervised learning is a very vast literature, with many methods and tricks developed over the years. The paper doesn't compare to any baselines developed for semi-supervised learning. Also, two of three experiments are with a simple linear model, so I'm guessing the pseudo-labels can't be that good. It would be convincing to see a comparison with state-of-the-art semi-supervised learning methods, with state-of-the-art predictive models, and with challenging datasets. I was also wondering why there was no comparison to Zhu et al. [14], since that paper operates in the same setting?

See "questions" for further comments.

---

> ### Author Rebuttal · Authors · 2025-07-30
>
> We appreciate the reviewer’s constructive feedback. It appears that the reviewer’s primary concern is a lack of experimental results.
> > My main concern is about whether the empirical evaluation is sufficient
>
> The main manuscript includes synthetic regression experiments, and two real data regression experiments—tabular and visual. While we believe the paper offers a reasonable balance between theory and empirical evaluation, we agree that additional classification experiments could strengthen our work, a comment that was also raised by reviewers sxXE and Gh11.
>
> *New image classification experiments:* In response, we have added new image classification experiments and a new baseline method: the approach of Zhu et al. [4] (Doubly-Robust Self-Training, DR-ST). We kindly refer the reviewer to response C4 to Reviewer sxXE for the details and empirical results. In short, to demonstrate the importance of the unbiased loss, we study the performance of teacher-student learning on CIFAR-10 under image corruption, termed CIFAR-10-C, which allows us to control the accuracy of the teacher model for the minority group. These experiments are in line with all other experiments presented in the paper; our PP-SSL method outperforms the standard baseline methods to which we compare—Teacher, OnlyLabeled, Standard SSL, PPI++, and DR-ST [4]. Below, we elaborate on all baseline methods we used in paper.
>
> *New experiments validating our key theoretical results:*
> - Specifically, we empirically verify Corollary 3.3 by showing that our theoretical $\lambda^*$ (the weight of the pseudo-labeled losses) aligns with the minimum of the gradient variance; kindly refer to response C3 to Reviewer NZKV.
> - We also provided a new experiment, showing that our algorithm for online learning λ successfully reaches the optimal λ; kindly refer to response C3 to Reviewer sxXE.
> ## C1: Paper doesn't compare to any baselines developed for SSL; what the baseline methods are and what is the teacher model?
>
> Thank you for the opportunity to clarify this point. We compared our method with common and simple SSL baselines as well as their PPI variants as follows.
>
> - **Teacher:** The teacher model $f$ is used for pseudo-labeling or for distillation, as widely used in the literature. The teacher is used to generate pseudo labels for the different teacher-student schemes, as we detail below.
> - **Standard SSL:** A model $g(x)$ trained on the following loss function: $\mathcal{L}_{Labeled}(g(X),y) + \mathcal{L}\_{Pseudo}(g(\tilde{X}),f(\tilde{X}))$, where $(X,y)$ are the labeled data, $\tilde{X}$ are the unlabeled data, and $f(\tilde{X})$ are the pseudo labels for the unlabeled data. For classification, $\mathcal{L}\_{Labeled}$ is the cross-entropy loss, and $\mathcal{L}\_{Pseudo}$ is the KL divergence between the class probabilities (softmax) of the teacher $f(\cdot)$ and student $g(\cdot)$ models. For regression, the two losses are the standard squared loss. As such, the Standard SSL model acts as the most widely used approach for semi-supervised learning.
> - **PPI++:**  A model $g(x)$ is trained using loss based on the PPI loss (Eq. (4)): $ \mathcal{L}\_{Labeled}(g(X),y) + \lambda \cdot (\mathcal{L}\_{Pseudo}(g(\tilde{X}),f(\tilde{X})) - \mathcal{L}\_{Pseudo}(g(X),f(X)))$. Notice that we use pseudo labels $f(X)$ for the labeled data as well. The value of λ is fixed and determined by the asymptotic variance minimization presented in PPI++[3]. We refer to this fixed λ as $\lambda\_{PP}^*$.
> - **Doubly-Robust Fixed-Teacher:** Variant of DR-ST [4] where the model $g(x)$ is also trained using the same loss function as PPI++ but has a heuristic for choosing the value of λ. Instead of a fixed value (as in PPI and PPI++), λ is set to 0 throughout training and switch it to 1 only during the final epoch as done in [4].
>  - **PP-SSL (Ours):** A model $g(X)$ based on the PPI++ loss but with an adaptive, online learning of λ. That is, the tunable parameter $\lambda_t$ is rigorously updated throughout the optimization steps, following our theory and the corresponding algorithm we proposed. The loss for step t becomes $ \mathcal{L}\_{Labeled}(g(X),y) + \lambda_t \cdot (\mathcal{L}\_{Pseudo}(g(\tilde{X}),f(\tilde{X})) - \mathcal{L}\_{Pseudo}(g(X),f(X)))$.
> - **OnlyLabeled:**  Model $g(x)$ trained only on the labeled data, for example, using cross-entropy loss (classification) or squared loss (regression). This is an important baseline as we want to show that using unlabeled data outperforms the OnlyLabeled model.
> ## C2:  What is $\lambda\_{PP}^*$, seeing that your gradients depend on the current iterate, unlike in PPI++ where there is a single distribution?
>
> As detailed above, $\lambda\_{PP}^*$ is a fixed λ, derived in PPI++ [3], which minimizes the asymptotic variance of the parameters of a linear model (more details are in [3, Section 6.1, Proposition 2]). In contrast, our theoretically-grounded method derives (i) the optimal λ for gradient variance minimization; and (ii) an online learning approach that automatically balances labeled and pseudo-labeled data contributions without requiring prior knowledge of variance parameters. Such an adaptive adjustment of λ does not exist in prior work.
> ## C3: Comparison to state-of-the-art baseline methods.
> We emphasize that our approach is orthogonal to many common SSL techniques, and they can be used in conjunction with our method. The goal of the paper is not to provide a SOTA SSL method; rather, we provide a theoretically-grounded framework for unbiased SSL.
>
> With that said, SOTA SSL methods can be modified to use our proposed unbiased loss, which provides a safeguard against undesired bias for minority groups for which teacher pseudo labels are less accurate, as explained below.
>
> Most state-of-the-art semi-supervised learning losses that use pseudo labeling can be written in a format of $\mathcal{L} = \mathcal{L}\_{Labeled}(g(X),y) + \lambda\cdot\mathcal{L}\_{Pseudo}(g(\tilde{X}),f(\tilde{X}))$. Therefore, to use both the SOTA SSL loss and our PP-SSL for unbiasedness, one can just update the loss to the form of: $\mathcal{L}\_{PP-SSL} = \mathcal{L}\_{Labeled}(g(X),y) + \lambda\cdot(\mathcal{L}\_{Pseudo}(g(\tilde{X}),f(\tilde{X})) - \mathcal{L}\_{Pseudo}(g(X),f(X)) )$.
>
> In our new experiment of CIFAR-10 classification (see Reviewer sxXE (C4)), we used the exact mechanism of combining SOTA SSL loss and our PP-SSL method. In that case, the pseudo loss, which is the KL divergence loss, is more complex than the standard loss functions used in previous experiments provided in the paper.
> ## C4: Why was there no comparison to Zhu et al., since that paper operates in the same setting?
> DR-ST [4] operates in a different setting; they keep updating the teacher as the training progresses, while we assume the teacher is fixed. Note that (1) the setting we consider is just as important as the self-training one, and (2) it is unclear how to provide in-depth theory for self-training for the reasons we share with reviewer sxXE, see our reply C1 (we target for theoretically-grounded algorithm for SSL).
>
> Indeed, DR-ST [4] do not provide any theory that justifies their choice for the update of λ, which they set to be 0 for all epochs and 1 for the last epoch. Also, they do not provide theoretical analysis for cases where the teacher is continuously updated.
> Following your question, we added an experiment with DR-FT variant of DR-ST where the teacher is fixed as done in all other baselines, (see Reviewer sxXE(C4)).  Using DR-ST heuristic of choosing λ. The results of this experiment show that DR-FT is comparable to PPI++ method, and our PP-SSL achieves higher accuracy (both overall and per group).
> ## C5: Technical questions
> > Factor of 4:
>
> Yes, by being asymptotically equivalent we mean the same dependence on n (and $\sigma^2$). In other words, it can be written as $\mathcal{O}(\sigma^2/n)$. The 4 factor comes from the inequality: $\lVert a+b+c+d\rVert^2 \leq 4\lVert a\rVert^2 + 4\lVert b\rVert^2 + 4\lVert c\rVert^2 + 4\lVert d\rVert^2$, which is used to prove Lemma 3.2 in Appendix B.1.
> > Correctness and bias:
>
>  By “correctness” we mean unbiasedness; we will revise the text to avoid any ambiguity. We care about unbiasedness because the loss in Eq. (4) is an empirical loss defined over a **finite sample** and our goal is to provide convergence guarantees for the **expected loss** over the distribution, as defined between lines 110 and 111. When optimizing with a biased gradient estimate:
> Empirically, we get worse performance where the teacher’s labels are incorrect (see the Standard-SSL results in our experiments, Section 4 and Appendices D,E,F.), and
> Theoretically, we cannot guarantee convergence to a stationary point, but only to its neighborhood, whose size depends on the bias (see Theorem 4 in [1]). In other words, the Standard-SSL baseline performance is highly affected by the teacher’s errors.
> > Using AdaGrad:
>
> As we mentioned in lines 223-227, our theoretical analysis relies on AdaGrad for both the model weights $w$ and the tuning parameter λ to properly adapt to $\sigma^2$ and $\sigma_e^2$, which we do not know in advance. AdaGrad provides a clear and simple second-order regret bound (Lemma 3.4), and it was commonly analyzed in previous work [2].
> Importantly, these second-order regret bounds (i.e., where regret depends on the sum of square gradients rather than their bounds) is crucial for our analysis. Nevertheless, In practice, any stochastic optimization method for $w$ and online learning method for λ can be used, such as SGD and Adam, which we also employ in our experiments. We will clarify this point in the text.
> ## References:
> [1] On the Convergence of SGD with Biased Gradients. Ajalloeian et al., 2020.
>
> [2] AdaGrad stepsizes: Sharp convergence over nonconvex landscapes. Ward et al., 2018.
>
> [3] PPI++: Efficient prediction-powered inference. Angelopoulos et al., 2023.
>
> [4] Doubly-Robust Self-Training. Zhu et al., 2024

---

> ### Comment · Reviewer_yHkN · 2025-08-05
>
> Thank you for the detailed response.
>
> I still do not understand your response to C2. The formula you are pointing out in [3] has $\theta^\star$. In [3] and other related works, that $\theta^\star$ is approximated by taking a $\sqrt{n}$-consistent estimator (for example, the output of the "only labeled" method). What do you use in place of $\theta^\star$?
>
> It seems that you feel that a core contribution of this paper is providing theoretical guarantees for semi-supervised learning. In that case, I can agree that a lot of heuristics from semi-supervised learning do not have this level of rigor. I will revise my rating accordingly.

---

> > ### Author Response · Authors · 2025-08-07
> >
> > Thank you for your thoughtful and constructive feedback. We appreciate your recognition of the theoretical contribution of our work, and we are especially grateful that you are considering a revision of your score. A key motivation behind our work is to move beyond heuristic-based methods and build a rigorous foundation for prediction-powered semi-supervised learning.
> >
> > In our experiments with linear models (synthetic, tabular, classification), $\theta^\*$ corresponds to the vector of regression coefficients, and $\lambda$ is set to minimize the variance of the estimated coefficients (using the labeled dataset). For age estimation, we use a deep neural network and follow a similar strategy: we define $\theta^\*$ as the vector of the last layer’s weights. In practice, this allows us to apply the same formula for setting $\lambda$ as in the linear case. We chose this approach because it is unclear how to set $\lambda$ to minimize the variance of the entire model’s parameters.
> >
> > This highlights the novelty of our contribution: transitioning from unbiased semi-supervised estimation of parameters $\theta^\*$ with a $\lambda$ that minimizes the variance of the estimator (as done in PPI++) to unbiased semi-supervised **model training** with an online or adaptive $\lambda$ that minimizes the variance of the gradients.

---

> > > ### Comment · Reviewer_yHkN · 2025-08-07
> > >
> > > Thanks for clarifying. This is important to mention in the paper, in my opinion.

---

> > > > ### Author Response · Authors · 2025-08-07
> > > >
> > > > We agree, and we truly appreciate your feedback. We will incorporate all clarifications and new results into the final version of the paper. Thank you again for your valuable input.

---

### Official Review · Reviewer_sxXE · 2025-07-03

**Clarity:** 3
**Significance:** 4
**Originality:** 3
**Rating:** 5
**Confidence:** 3

**Summary:**

This paper proposes Prediction-Powered Semi-Supervised Learning (PP-SSL), which incorporates the bias-correction principle of Prediction-Powered Inference (PPI) into the training process of semi-supervised learning (SSL). The method addresses the bias introduced by inaccurate pseudo-labels using an unbiased gradient estimator, and dynamically integrates it into training via an online learning algorithm. Theoretical analysis establishes finite-time convergence guarantees, and empirical results on synthetic, tabular, and image-based regression tasks demonstrate that PP-SSL outperforms both classical SSL methods and previous PPI-style approaches under distributional shift.

**Questions:**

1. The explanation of the teacher model is not clear. If I understand correctly, the paper focuses exclusively on fixed teacher models. It is unclear how the method would adapt to self-training or evolving pseudo-label generators, which are more common in modern SSL pipelines. I would appreciate it if the authors could clarify this point.

2. While the online λ tuning is theoretically well-justified, there is limited qualitative analysis or visualization. Can the authors provide qualitative visualizations of λ values during training and how they correlate with pseudo-label error over time?

3. Are there comparisons between online λ and fixed λ values to better illustrate the benefits of adaptive tuning?

4. Have the authors considered extending PP-SSL to image classification tasks like CIFAR-10 or STL-10? The absence of popular benchmark datasets may limit direct comparison with standard SSL baselines. It would be helpful to demonstrate how PP-SSL behaves in more common classification settings.

**Ethical Concerns:**

["NO or VERY MINOR ethics concerns only"]

**Final Justification:**

The authors' rebuttal provides clear and thoughtful responses, supported by additional experiments on image classification and the behavior of the adaptive λ. These results sufficiently address the concerns raised in my initial review. I agree that establishing a theoretical foundation for unbiased semi-supervised learning via the PPI loss is a meaningful and worthwhile contribution. My overall evaluation remains positive, and I have kept my original score.

**Limitations:**

The authors assume that labeled and unlabeled data come from the same distribution, which can be a practical limitation. It would be helpful if the paper included a more discussion of this limitation.

**Paper Formatting Concerns:**

.

**Quality:**

4

**Strengths And Weaknesses:**

The paper presents an insightful extension of the Prediction-Powered Inference (PPI) framework from inference to optimization, specifically tailored for semi-supervised learning. Unlike prior work, it directly addresses the practical issue of pseudo-label bias during training and introduces a principled mechanism to mitigate it. The paper is generally well-organized, with clear motivation, formulation, and theoretical analysis.

Nevertheless, several questions remain regarding the generality of the datasets, the explanation of the teacher model, and the analysis of the learned λ. Please refer to the questions.

---

> ### Author Rebuttal · Authors · 2025-07-30
>
> We thank the reviewer for their thoughtful and constructive feedback. We appreciate the recognition of our work as a principled extension of Prediction-Powered Inference (PPI) to semi-supervised learning (SSL), and for acknowledging the clarity of our motivation, theoretical contributions, and empirical results. Below we address the reviewer’s main concerns:
>
>
> ## C1: How the method would adapt to self-training or evolving pseudo-label generators
> You are correct that our current theoretical analysis focuses on a fixed teacher model. It is a widely used learning strategy, and so we believe it is important to build a clear theoretical foundation for unbiased semi-supervised learning via the PPI loss. We agree that extending our method to the case of self-supervised learning with evolving pseudo-label generators is an important direction. Below, we explain what makes such an extension challenging.
>
>
> Our method assumes a pre-trained, fixed teacher model that generates pseudo-labels throughout training. This design choice allows us to (i) theoretically analyze the convergence and dynamics of the PPI loss as the pseudo-label error variance σₑ² remains constant, and (ii) leverage regret bounds for the optimal λ.
>
>
> *From a theoretical perspective:*
> If we assume the teacher evolves over time, $f_1,\ldots,f_t$, and assuming the teacher is independent of the training data (e.g., using held-out data), our results directly generalize with σₑ² as a bound over all teachers. However, this would yield loose bounds. A more refined theory  requires deriving new gradient variance bounds that depend on the evolving sequence of the teacher’s error gradient variance σₑₜ²; naturally, this is a more complex theoretical setup than the one considered in our paper.
> The common self-training setup, where the teacher evolves based on the training data (e.g., without using held-out data), is even more challenging because (i) it is unclear how to bound the gradient variance (due to the dependence on past data), and (ii) it requires moving from static to dynamic regret bounds as the optimal λ* changes over time (i.e., λₜ*); clearly, this transition is challenging and not trivial.
>
>
> *From a practical perspective:* While our theoretical guarantees are established for fixed teachers, our algorithm can be practically applied to self-training settings. In such cases, the same model serves as both student and teacher, with pseudo-labels being iteratively updated, and the PPI loss would still have the unbiasedness effect. Of course, this idea requires more exploration and validation that is beyond the scope of our current paper.
>
> ## C2 : Qualitative visualizations of λ values during training and how they correlate with pseudo-label error over time
> We appreciate this request for deeper insight into λ dynamics. It's important to note that in our fixed teacher setting, pseudo-label error remains constant over time since the teacher model is fixed. What could be correlated are the final λ values and the pseudo-label error, and so we provide such an analysis below.
> We conducted a synthetic linear regression experiment where both the data generation process and the performance of the teacher model are controlled (see Reviewer NKZV, Comment C3, for experimental details). We will add the visualizations to the revised version.
>
> The following table shows the value of $\lambda_t$ for different epochs for teachers with varying pseudo-label errors (MSE). We initialize the learning with $\lambda_0 = 1$. Observe how $\lambda_t$ adjusts automatically, reaching the theoretical $\lambda^\*$ from Corollary 3.3 across different runs. Observe also that $\lambda^\*$ is increasing as the teacher’s MSE decreases, in line with the intuition.
>
> | Teacher model MSE|\| $\lambda\_{0}$ | $\lambda\_{100}$ | $\lambda\_{200}$ | $\lambda\_{300}$ | $\lambda\_{400}$ | $\lambda\_{500}$ | $\lambda\_{600}$ | $\lambda\_{700}$ | $\lambda\_{800}$ | $\lambda\_{900}$ | $\lambda\_{1000}$ |\| $\lambda^*\_{\text{theory}}$ |
> |-|- |- |- |- |- |- |- |- |- |- |- |- |
> |0.01|\|  1.000 | 0.966 | 0.946 | 0.934 | 0.929 | 0.926 | 0.925 | 0.924 | 0.924 | 0.924| 0.924|\| 0.968 |
> |0.04|\| 1.000 | 0.945 | 0.902 | 0.866 | 0.836 | 0.811 | 0.791 | 0.775 | 0.761 | 0.751| 0.742 |\| 0.795 |
> |0.05|\| 1.000 | 0.930 | 0.867| 0.812 | 0.766 | 0.727 | 0.696 | 0.671 | 0.651 | 0.635| 0.622 |\| 0.634 |
>
>
> ## C3: Are there comparisons between adaptive λ and fixed λ?
>
>
> Yes, in the paper we compare our adaptive method to PPI++, which uses a fixed, asymptotically optimal value of λ. (We also compared in our experiments to PPI with λ=1, but we omitted the latter as we found its performance to be inferior to PPI++.) In nearly all experiments, our adaptive method outperforms PPI++.
>
>
> To fully address the reviewer's concern, we now include a new experiment that compares our method, PP-SSL, to PPI with an infeasible “best” λ that minimizes the error on the test data. We use the synthetic regression setup with subgroup bias, as described in Section 4.1. The results (see below) show that our adaptive algorithm matches the performance of the model trained using the best fixed λ across all bias settings.
>
> | Bias $\mu$ |\| Infeasible “best” λ | Infeasible “best” MSE (Total / A / B) |\| PP-SSL λ | PP-SSL MSE (Total / A / B) |
> |:--|--|--|-|--|
> | 0.1 |\| 0.50| 1.93 / 1.76 / 2.59 |\| 0.46| 1.93 / 1.77 / 2.58  |
> | 0.5 |\| 0.50| 1.85 / 1.69 / 2.50|\| 0.59 | 1.85 / 1.69 / 2.50  |
> | 3.0 |\| 0.35| 2.76 / 1.95 /6.05|\| 0.33| 2.76/ 1.96 / 6.04  |
> | 5.0 |\|0.35 | 2.45 / 2.07 / 4.05 |\| 0.36| 2.47/ 2.06 / 4.12|
>
> ## C4: Evaluation on image classification task
> Following your suggestion, we now include new results of PP-SSL for a semi-supervised image classification task. To illustrate the importance of the unbiased loss, we designed a controlled experiment on CIFAR-10 and its corrupted variant CIFAR-10-C [1]: we divided the data into 2 groups based on the classes, simulating a setting where the teacher is less accurate for some classes (Group A: dog, cat, and deer images from CIFAR-10-C) and more accurate for the other classes (Group B: images from the clean CIFAR-10). Image corruptions—Saturate, Spatter, and SpeckleNoise—were drawn from the CIFAR-10-C benchmark and applied to both training and test images of Group A.
>
> We followed a standard SSL protocol: a ResNet50 pretrained on ImageNet was used to extract features from all images, and a linear classifier was trained on top using 10% labeled data and 90% unlabeled data. Specifically, the benchmark methods are as follows.
> - Teacher: a model trained on a clean, disjoint labeled subset using cross-entropy loss.
> - Standard SSL: Student model trained using CE loss for labeled data and KL loss for unlabeled data (classic knowledge distillation setting).
> - OnlyLabeled: Student model trained using CE loss, evaluated on labeled data.
> - PPI++: Student model trained using CE loss for labeled data and KL loss for pseudo-labeled data (both unlabeled and labeled as in Eq. (4)), with a fixed weight λ for the PPI loss.
> - Doubly-Robust Fixed-Teacher (DR-FT): Variant of Doubly-Robust Self-Training [2], that follows PPI++ loss but with heuristic update of λ, which is set to 0 throughout training and switched to 1 only during the final epoch.
> - Ours PP-SSL: Same as PPI++ but using our adaptive λ algorithm rather than a fixed one (PPI++) or a heuristic update (DR-FT).
>
> Following the table below, we can see that PP-SSL consistently outperforms all baselines in terms of total accuracy and Group A (corrupted) accuracy, while maintaining comparable performance on Group B (clean) samples. Notably, the performance gap on Group A varies with the corruption type, reflecting the varying inaccuracies of the pseudo-labeling under different distortions. To ensure the generality of these findings, we also ran the same experiment across other corruption types and severity levels. Since the results exhibited similar trends, we omit them for clarity and focus here only on severity level 5.
> | Method||Saturate|||Spatter||| Speckle Noise ||
> |:--|:-|-|-|:-|-|-|:-|-|-|
> | *Accuracy*| *Total*|*Group A* | *Group B* | *Total*|*Group A*| *Group B* | *Total*|*Group A*  | *Group B*|
> | DR-FT | 79.83±0.17 | 67.65±0.35 | 85.05±0.18 | 76.50±0.09 | 55.67±0.32 | 85.43±0.16 | 78.47±0.17 | 62.26±0.34 | 85.42±0.16 |
> | Labeled | 80.76±0.13 | 63.05±0.42 | 88.35±0.03 | 66.06±0.71 | 13.81±2.38 | 88.46±0.04 | 77.82±0.38 | 53.02±1.25 | 88.44±0.05 |
> | SSL | 80.67±0.15| 62.61±0.47| 88.41±0.02 | 66.25±0.70 | 14.15±2.36 | 88.57±0.03 | 77.95±0.37 | 53.25±1.23 | 88.53±0.03|
> | PP-SSL (Ours)| **83.31±0.02**| **72.94±0.11** | 87.76±0.04| **79.20±0.13** | **58.07±0.40** | 88.26±0.05 | **81.52±0.13** | **65.76±0.36** | 88.27±0.05 |
> | PPI++ |79.96±0.08 | 68.01±0.2 | 85.08±0.09 | 76.75±0.08 | 56.37±0.38 |85.49±0.09 | 78.51±0.12 | 62.36±0.34 | 85.44±0.09 |
> | Teacher| 76.92±0.12| 49.88±0.35 | **88.51±0.05** | 65.00±0.18 |10.01±0.67 | **88.57±0.04** | 76.12±0.11 | 47.16±0.35 | **88.54±0.05**|
> ## C5: Assumption that labeled and unlabeled data come from the same distribution.
> Thank you for raising this important point, which was also noted by Reviewer Gh11 (see Comment C2). As discussed there, while PP-SSL assumes a shared distribution between labeled and unlabeled data, we believe the framework can be extended to address distribution shift—for example, through importance weighting of loss or gradients, as in prior work (e.g., PPI, Zhu et al., 2023). These ideas could be incorporated into our adaptive λ scheme. We also note that in more complex cases—such as using generative models to produce unlabeled data—additional adaptations to the pseudo-labeling and gradient-weighting components of PP-SSL may be necessary. We will expand on these directions in the final version of the discussion section.
>
> ## References:
> [1] Benchmarking Neural Network Robustness to Common Corruptions and Perturbations. Hendrycks et al., 2019.
>
> [2] Doubly-Robust Self-Training. Zhu et al.,NeurIPS 2024

---

> > ### Comment · Reviewer_sxXE · 2025-08-05
> >
> > Thank you for the detailed response. The additional experiments on image classification, along with the experiments related to λ, have addressed my concerns. While the method involves certain assumptions, such as the use of a teacher model and specific distributional settings, I agree with the authors that establishing a clear theoretical foundation for unbiased semi-supervised learning via the PPI loss is an important and valuable direction. My overall assessment remains positive, and I have decided to keep my score unchanged.

---

> > > ### Author Response · Authors · 2025-08-07
> > >
> > > Thank you for your positive assessment and thoughtful feedback. We're glad the additional experiments and discussion addressed your concerns, and we appreciate your recognition of the value in establishing a theoretical foundation for unbiased semi-supervised learning. We will incorporate the suggested clarifications and additions into the final version of the paper.

---

### Official Review · Reviewer_NZKV · 2025-07-03

**Clarity:** 3
**Significance:** 3
**Originality:** 2
**Rating:** 4
**Confidence:** 2

**Summary:**

The paper proposes PPI-SSL, which extends the Prediction-Powered Inference (PPI) framework to the semi-supervised learning (SSL) setting. It introduces a way to balance the contributions of labeled and pseudo-labeled data by adaptively tuning a weighting parameter $\lambda$, updated via AdaGrad.

**Questions:**

- Please clarify what is truly novel in the paper beyond the AdaGrad update of \lambda.
- How realistic is the Lipschitz assumption in Lemma 3.1, and how is it enforced in practice?
- Could you verify Corollary 3.3 empirically on a synthetic setting, and is it used to initialize \lambda in real experiments?

**Ethical Concerns:**

["NO or VERY MINOR ethics concerns only"]

**Final Justification:**

I thank the authors for the extra clarification and synthetic experiment. I am raising my score as my concerns have been addressed.

**Limitations:**

Limitations are discussed in the paper, and there are no obvious societal concerns.

**Paper Formatting Concerns:**

no formatting concern

**Quality:**

3

**Strengths And Weaknesses:**

## Strength
- The paper is clearly written and well organized.
- The proposed adaptive mechanism for $\lambda$ is intuitive and practically motivated.

## Weakness
- I am not an expert in this area and not a theory person, but as far as I understand, the formulation of Eq. (3) has already appeared in prior works like PPI and Doubly-Robust Self-Training. So it is not accurate to say this paper newly “extends its core idea to semi-supervised learning (SSL) for model training.” At least that part is not new. The relatively novel contribution seems to be the AdaGrad-based update of $\lambda$, which deserves clearer emphasis.
- Lemma 3.1 assumes the gradient function is Lipschitz, which seems like a very strong assumption. I’m wondering how realistic this is for modern networks, and whether any regularization is applied to enforce it. (In practice, it’s more common to assume the model is Lipschitz rather than the gradient itself.)
- It would be helpful to empirically verify Corollary 3.3 in a toy experiment, e.g., using a simple MLP where the bound involving $\sigma$ can be computed. Also, is this bound used in practice to initialize $\lambda$? If so, please clarify.

---

> ### Author Rebuttal · Authors · 2025-07-30
>
> Thank you for your thoughtful review and constructive feedback. We appreciate your careful evaluation of our work and the opportunity to address your concerns.
>
> We understand that your major concern is with the novelty of our work, so we start by clarifying this issue and then proceed by responding to the rest of your comments.
>
> ## C1: Novel Contributions Beyond AdaGrad Update;  the formulation of Eq. (3)
>
> You correctly note that Eq. (3), which we refer to as the PPI loss in the paper, has appeared in prior works. It was introduced in PPI [1], and used in PPI++ [2], and Doubly-Robust Self-Training [3]. However, our work introduces several key distinctions that extend beyond simply applying AdaGrad to update the tuning parameter λ of the PPI loss that weighs the influence of the unlabeled data.
>
> First, to clarify, PPI and PPI++ focus on constructing prediction intervals for parameters of interest, such as the mean or quantile of a parametric distribution. These two papers do not utilize the unbiased loss for *model training*. As such, they differ from the focus of this paper, which centers on the formulation of an unbiased risk for semi-supervised learning. The Doubly Robust Self-Training [3] method does involve model training; however, our work introduces key novelties in relation to [3], as well as to [1,2], which we outline in lines 32-39 and summarize here:
>
> *Shift from Asymptotic to Finite-time Analysis:* While PPI, PPI++, and Doubly-Robust Self-Training analyze only the asymptotic behavior of Eqs. (3)-(4) at the **population optimum**, our work focuses on finite-time analysis during training. Specifically, we explicitly quantify the benefit of the teacher depending on its quality (teacher error), and show how and when it enables better generalization guarantees compared to the standard case. Such a finite-time, dynamic analysis of the optimization process does not exist in prior work.
>
> *Adaptive Training Mechanism:* Our second main contribution is the adaptive adjustment of λ during training, which automatically balances labeled and pseudo-labeled data contributions without requiring prior knowledge of variance parameters. This eliminates manual tuning and provides greater robustness compared to fixed-weight approaches. Such an adaptive adjustment of λ does not exist in prior work.
>
> *Performance Gain:* Our method demonstrates superior performance under group imbalance, where the teacher model performs well on one group but underperforms on the other. In such cases, traditional SSL methods fail, and existing PPI-based methods underperform compared to our method, despite using the same loss function but with a fixed lambda.
>
> ## C2:  How realistic is the Lipschitz assumption in Lemma 3.1, and how is it enforced in practice?
>
> The Lipschitz assumption in Lemma 3.1 applies to the *loss function*, not the model itself. This is an important distinction that makes the assumption more realistic in practice.
>
> *Why is it realistic?* We show in Appendix A that this assumption naturally holds for the squared loss (regression) and the logistic loss (classification). The assumption provides the theoretical foundation for relating the loss error gradient variance σₑ² to the teacher's prediction error $\mathcal{E}^f$.
>
> *Importantly, we do not need to enforce this assumption.*  Our analysis holds under the variance conditions specified in Eqs. (8) and (9): bounded gradient variance σ² and bounded loss error gradient variance σₑ². The Lipschitz assumption serves only to provide intuition and theoretical interpretation of how the teacher's prediction error relates to the loss error gradient variance. In other words, our theory relies on the assumptions in Eqs. (8) and (9) and holds even if the loss gradient $\nabla\ell(w;x,y)$ is not Lipschitz in $y$. Therefore, there is no need to enforce the Lipschitz assumption in practice.
>
>
> ## C3: Verifying Corollary 3.3 and λ initialization in our experiments
>
> Thank you for raising this important point. **No**, the theoretical λ* from Corollary 3.3 is not used for initialization since it depends on the unknown quantities σ² and σₑ². Instead, we simply initialize it to $\lambda_0=1$. Our adaptive mechanism updates λ during training without requiring these parameters. We also provide a new experiment, showing that our algorithm for online learning λ aligns with the optimal $\lambda^*$; we kindly refer the reviewer to response C3 to Reviewer sxXE.
>
> Turning to the empirical validation of Corollary 3.3, we have conducted a new synthetic regression experiment to validate the theoretical λ* from Corollary 3.3 against the empirical λ that minimizes the gradient variance. We emphasize that $\lambda^*$ from Eq. (10) is the minimizer of the variance's **upper bound**, which might differ from the minimizer of the variance itself, depending on how loose the upper bound is for a specific problem setting. See results and full experimental setup below.
>
> *Results:* The table below shows that $\lambda^\*$ from Corollary 3.3 (denoted as $\lambda^\*\_{\text{theory}}$) attains nearly the same performance, in terms of gradient variance (**Var($\cdot$)**), as the empirical $\lambda^*\_{\text{practice}}$, for different values of $r,\sigma^2,\sigma_e^2$.
>
> |      r   |      σ² |     σₑ² |   $\lambda^*\_{\text{theory}}$ (Cor. 3.3) |  $\lambda^*\_{\text{practice}}$ (Empirical) |  Var($\lambda^*\_{\text{theory}}$) |   Var($\lambda^*\_{\text{practice}}$) |
> |:--------:|:--------:|:--------:|:------------------------:|:------------------------:|:-----------------:|:-------------------:|
> | 0.025 | 10.4078 |  0.0793 | 0.9682 |      0.99 |           0.0050 |             0.0046 |
> | 0.025 | 14.1715 |  7.6479 | 0.6336 |     0.70 |           0.1733 |             0.1669 |
> | 0.25  | 11.5082 |  0.0440  | 0.7969 |     0.77 |           0.0749 |             0.0709 |
> | 0.25  | 11.9349 | 10.1486 | 0.4324 |      0.35 |           0.1034 |             0.1004 |
>
>
> *Experimental setup:*  We design a synthetic experiment to analyze the behavior of prediction-powered gradients in a controlled linear regression setting. The data is generated according to the model $y = \mathbf{x}^\top \mathbf{w}^* + \epsilon$, where $\mathbf{x} \sim \mathcal{N}(0, I_d)$, $\epsilon \sim \mathcal{N}(0, \sigma_*^2)$ with $\sigma_*^2=0.01$, and $\mathbf{w}^* \in \mathbb{R}^d$ is a parameter vector sampled once per configuration from $\mathcal{N}(0,I_d)$. We sample $n=50$ labeled samples from this model and $N\in$ {$200, 2000$} unlabeled samples from the same feature distribution $\mathcal{N}(0, I_d)$. A synthetic teacher provides pseudo-labels of the form $f(\mathbf{x}) = \mathbf{x}^\top (\mathbf{w}^* + b \mathbf{e}_1) + \zeta$, where $\mathbf{e}_1 = [1,0,0,...,0]$, $b\in\mathbb{R}$ controls the teacher bias, and $\zeta\sim\mathcal{N}(0, \sigma\_{\zeta}^2)$. This setup gives control over $\sigma_e^2$ through the parameters $\sigma\_{\zeta}^2$ and $b$. The results in the table correspond to a randomly sampled test point $\tilde{w}\sim\mathcal{N}(0,I_d)$.
>
> ## References:
> [1] Anastasios N Angelopoulos, Stephen Bates, Clara Fannjiang, Michael I Jordan, and Tijana
> Zrnic. Prediction-powered inference. Science, 2023.
>
> [2] Anastasios N Angelopoulos, John C Duchi, and Tijana Zrnic. PPI++: Efficient prediction-
> powered inference. arXiv preprint arXiv:2311.01453, 2023.
>
> [3] Banghua Zhu, Mingyu Ding, Philip Jacobson, Ming Wu, Wei Zhan, Michael Jordan, and Jiantao Jiao. Doubly-Robust Self-Training. Advances in Neural Information Processing Systems, 2024

---

> > ### Author Response · Authors · 2025-08-07
> >
> > We hope our rebuttal has addressed your concerns. We’d be grateful to hear from you if you have any follow-up questions or additional feedback.

---

### Note · Authors · 2025-08-13

We appreciate the reviewers’ overall positive assessment of our work, acknowledging that leveraging online power tuning in a learning context “is a great direction” (Reviewer yHkN) and that “establishing a clear theoretical foundation for unbiased semi-supervised learning via the PPI loss is an important and valuable direction” (Reviewer sxXE). We have fully addressed all reviewers’ concerns and couldn't have summarized the rebuttal better than Reviewer Gh11: “Overall, your clarifications, added experiments, and forward-looking extensions substantially bolster the paper”.

It appears that the main concern raised by the reviewers was the sufficiency of our empirical evaluation. In response, we have added image classification experiments on the CIFAR-10 dataset under corruption (CIFAR-10-C). In addition, we have included visualizations of the tuning parameter λ over time, along with experiments validating its relation to the theoretical values, which provide insights into its training dynamics and alignment with the theory. To quote Reviewer sxXE: “The additional experiments on image classification, along with the experiments related to λ, have addressed my concerns”; and Reviewer Gh11: “I also appreciate the additional empirical validations, especially the extension to widely used benchmarks like CIFAR-10 and CIFAR-10-C. I think they further strengthen the practical significance of your contributions.”

We enjoyed the valuable discussion with the reviewers and their forward-looking perspectives on future research directions. In this work, we have considered the widely used teacher-student setup and focused on establishing a fundamental convergence theory for unbiased SSL. As we mentioned in the paper, extending our work to the self-training setting (where the teacher evolves over time, which is beyond our current scope) is a promising direction. Another exciting avenue is handling distributional shift. Our discussion with the reviewers helped us further explore these directions more concretely (“the ideas of importance weighting, reweighting gradient estimators, and the proposed double-loop iterative scheme are concrete and promising directions” (Reviewer Gh11)), and we will revise our discussion accordingly.

We sincerely thank the reviewers for their comments and helpful suggestions. We will incorporate all suggestions into the final version, including the added experiments, and further elaborate on future research directions.

---

### Decision · Program_Chairs · 2025-09-17

**Decision:**

Accept (poster)

**Comment:**

This paper introduces PP-SSL, which brings the prediction-powered inference (PPI) debiasing principle into semi-supervised training by using an unbiased gradient estimator combined with online tuning of the interpolation weight lambda (via an AdaGrad-style update), yielding finite-time convergence and variance-reduction guarantees. Main commendations include principled extension of PPI to SSL training with theory (sxXE, Gh11, and yHkN), the idea of adaptive online tuning of lambda is novel and sensible (yHkN and Gh11), empirical strength and added image experiments (sxXE and Gh11, post-rebuttal). Main concerns include assumption on distributions (sxXE and Gh11). Overall, I think this is a nice paper that deserves to be accepted.